# Anatomy of a fumarole field; drone remote sensing and petrological approaches reveal the degassing and alteration structure at La Fossa cone, Vulcano Island, Italy

Daniel Müller[1], Thomas R. Walter[1], Valentin R. Troll[2,3], Jessica Stammeier[1], Andreas Karlsson[4], Erica de Paolo[5], Antonino Fabio Pisciotta[6], Martin Zimmer[1], Benjamin De Jarnatt[1]

[1] GFZ German Research Centre for Geosciences, Telegrafenberg, 14473 Potsdam, Germany
[2] Dept. of Earth Sciences, Natural Resources and Sustainable Development, Uppsala University, Sweden
[3] Istituto Nazionale di Geofisica e Vulcanologia (INGV), Rome, Italy
[4] Department of Geosciences, Swedish Museum of Natural History, Box 50007, SE-104 05 Stockholm, Sweden
[5] University of Milano-Bicocca, Department of Earth and Environmental Sciences, Piazza della Sciencza 4 – 20126 Milano, Italy
[6] Istituto Nazionale di Geofisica e Vulcanologia (INGV), Palermo, Italy

*Correspondence to*: Daniel Müller (dmueller@gfz-potsdam.de)

**Abstract.** Hydrothermal alteration and mineralization processes can affect the physical and chemical properties of volcanic rocks. Aggressive acidic degassing and fluid flow often also leads to changes in the appearance of a rock, such as changes in surface coloration or intense bleaching. Although hydrothermal alteration can have far-reaching consequences for rock stability and permeability, yet limited knowledge exists on the detailed structures, extent, and dynamic changes that take place near the surface of hydrothermal venting systems. By integrating drone-based photogrammetry with mineralogical and chemical analyses of rock samples and surface gas flux, we investigate the structure of the evolving volcanic degassing and alteration system at the La Fossa cone on the island of Vulcano, Italy. Our image analysis combines Principal Component Analysis (PCA) with image classification and thermal analysis, through which we identify an area of approximately 70,000 m² that outlines the maximum extent of hydrothermal alteration effects at the surface, represented by a shift in rock color from reddish to gray. Within this area, we identify distinct gradients of surface coloration and temperature that indicate a local variability of degassing and alteration intensity and define several structural units within the fumarole field. At least seven of such larger units of increased activity could be constrained. Through mineralogical and geochemical analysis of samples from the different alteration units, we define a relationship between surface appearance in drone imagery and the mineralogical and chemical composition. Gradients in surface color from reddish to gray correlate with a reduction of $Fe_2O_3$ from up to 3.2% in the unaltered regime to 0.3% in the altered regime, and the latter coincides with the area of increased diffuse acid gas flux. As the pixel brightness increases towards higher alteration gradients, we note a loss of the initial (igneous) mineral fraction and a change in bulk chemical composition with a concomitant increase in sulfur content from close to 0% in the unaltered samples to up to 60% in samples from the altered domains. Using this approach of combined remote sensing and in situ analyses, we define and spatially constrain several alteration units and compare them to the present-day thermally active surface and degassing pattern over the main crater area. The combined results permit us to present a detailed anatomy of the La Fossa fumarole field, including high-temperature fumaroles and seven larger units of increased alteration intensity, surface temperature, and variably intense surface degassing. Importantly, we also identify apparently sealed surface domains that prevent degassing, likely as a consequence of mineral precipitation from degassing and alteration processes. By assessing the thermal energy release of the identified spatial units quantitatively, we show that thermal radiation of high-temperature fumaroles accounts for < 50% of the total thermal energy release only, and that the larger part is emitted by diffuse degassing units. The integrated use of methods presented here has proven to be a useful combination for a detailed characterisation of alteration and activity patterns of volcanic degassing sites and has potential for application in alteration research and for monitoring of volcanic degassing systems.

## 1 Introduction

### 1.1 Volcanic degassing and hydrothermal alteration

Volcanic degassing at the Earth's surface is typically expressed in the form of localized fumarole fields and diffuse degassing. Yet, the association of localized and diffuse degassing is not well constrained. A fumarole is a vent or opening in the Earth's surface that releases steam and gas, including sulfur dioxide, carbon dioxide, and hydrogen sulfide, into the atmosphere (e.g. Giggenbach, 1996; Giammanco et al., 1998; Halldorsson et al., 2013). Fumaroles are typically found near volcanic areas or geothermal regions where there is intense heat beneath the surface. Fumaroles are of interest to scientists studying volcanoes and geothermal systems, as they provide information on the composition of underlying magmatic systems, the degassing processes of such magmatic systems, and the dynamic changes in the degassing passways exploited by such systems (e.g. Chiodini et al., 1993; Aiuppa et al., 2005; Paonita et al., 2013). The gas emissions by fumaroles, moreover, provide information on the possible interaction between underground water and hot rocks or magma and thus the state of a hydrothermal system through time (e.g. Chiodini et al., 1993; Capasso et al., 2000; Nuccio et al., 2001; Troll et al., 2012; Paonita et al., 2013).

The degassing of hot and acid volcanic gasses leads to versatile fluid-rock interactions at the surrounding volcanic rock, summarized as hydrothermal alteration (Pirajno, 2009; Chiodini et al., 2013; Fulignati, 2020). Alteration can affect the mineral assemblage by dissolution and remineralization up to complete destruction of the original mineral matrix and eventually influence essential rock parameters with potentially far-reaching consequences for the shallow hydrothermal system and the stability of a volcanic building (Reid and Brien, 2001; Heap and Violay, 2021). Mechanical strength tests of hydrothermally altered rocks showed considerable mechanical weakening (e.g. Frolova et al., 2014; Heap et al., 2021a, Darmawan et al., 2022), which is usually accomplished by mineral dissolution and mineral re-precipitation that affect rock strength and permeability and can in cases even seal gas pathways. Hydrothermal alteration can thus lead to sealed rock masses and hence to pressure build-up in a shallow volcanic system and consequently influence volcanic activity (e.g. Heap et al., 2019) and increase the likelihood of flank deformation and collapse (Heap et al., 2021b). It is therefore important to better understand the degassing and alteration structures in active hydrothermal crater regions of hazardous volcanic systems.

In this study of the fumaroles of La Fossa cone, Vulcano Island - Italy, we aim to detect and quantify alteration-related spatial and compositional parameters in order to provide improved insight into the dynamic changes of hydrothermal venting systems to help identify temporal and potentially critical developments and to better understand the associated features of diffuse and localized degassing.

### 1.2 Structure and extent of degassing sites

Recent advances in volcanic geothermal areas suggest that fumaroles are often only localized expressions of a much larger area of degassing (e.g. Toutain et al., 2009; Liuzzo et al., 2015). Indeed, fumaroles and hydrothermal degassing zones are often accompanied by broader fields of activity, characterized by diffuse degassing processes, associated mineral changes, and

intense surface recoloration (e.g. Donoghue et al., 2008; Berg et al., 2018; Darmawan et al., 2022) and fumaroles activity can vary in time (Troll et al., 2012; Fischer et al., 2015) and in size (Lynch et al., 2013; Gertisser et al 2023). Previous works at Vulcano, for instance, have shown that fumaroles are surrounded by extensive areas of diffuse degassing (Carapezza et al., 2011; Chiodini et al., 2005; Manini et al., 2019). Our previous work showed that diffuse degassing leads to distinct zones of activity that can be identified by temperatures and visual expression (Müller et al., 2021). Those diffuse zones are typically constrained based on $CO_2$ measurements but are also subject to the diffuse flow of acid gas driving diffuse alteration processes. However, these diffuse degassing and alteration processes are often difficult to recognize without specialized sampling strategies (Toutain et al., 2009), leading to a limited understanding of the anatomy and extent of degassing and alteration systems. Understanding the dynamic changes and internal architecture of hydrothermal activity of fumarole fields and the true dimensions of their field of activity is of relevance for the study of volcanic activity and hazard potential.

## 1.3 Surface effects and remote sensing of alteration

Hydrothermal alteration can cause significant changes in the physical and chemical properties of volcanic rock, such as density, compressive strength, and permeability (e.g. Donoghue et al., 2008, 2010; Berg et al., 2018; Heap et al., 2019; Darmawan et al., 2022). The replacement of primary minerals by secondary minerals, element mobility of fluid-mobile components, enrichment of refractory elements, and physical and textural changes of rock properties are often accompanied by changes in the color or spectral reflectance characteristics and can be traced employing remote sensing techniques.

Several studies have investigated the relationship between coloration and hydrothermal alteration. The use of rock color or spectral reflectance particularities as an indicator of alteration has been explored since the 1970s and led to the development of a variety of satellite remote sensing techniques using ETRS multispectral imagery (Rowan et. al., 1976), Landsat Thematic Mapper mission (Carranza et al., 2002), ETM+ (Mia et al., 2012), ASTER data (Di Tommaso et al., 2007) or hyperspectral data (e.g. Van De Meer et al., 2012; Tayebi et al., 2015). These techniques can detect subtle changes in color that may not be visible to the naked eye, allowing for the identification of mineral deposits (Mielke et al., 2016), hydrothermal alteration, or volcano stability (Kereszturi et al., 2020).

However, for analyzing details of localized degassing and alteration systems, the resolution of satellite data is often a limiting factor. Some of the best available optical satellite data have a resolution of 0.5 m in the nadir acquisition position. The resolution of thermal satellite data is on the order of tens to hundreds of meters per pixel. That allows the general detection of degassing and alteration systems, but the imaging of details of such systems requires the use of very high-resolution data. Modern UAS (unmanned aerial systems) equipped with high-resolution sensors allow imaging of volcanic surfaces at cm scales and, therefore, permit the analysis of degassing and alteration systems in great detail. In combination with Structure from Motion (SfM) processing, they are efficient for first-site investigations and allow the creation of high-resolution structural maps to identify structures of degassing systems to assist first-order hazard analysis or guide further in-depth studies.

## 1.4 Aim of the study

The aim of this work is to image and analyze the degassing and alteration structure of the La Fossa fumarole field and the wider field of activity and better understand the association of diffuse and localized degassing and alteration at degassing sites. We advance previous results (Müller et al., 2021) by considering new data, and by integrating them with the mineralogical and chemical analysis of alteration distribution in collected rock samples. We show systematic changes in the effects of alteration on the surface coloration and how drone-derived RGB data (Red, Green, Blue, standard color coding of images) can be used for the efficient detection and classification of degassing and alteration features. Combining UAS-based optical and infrared remote sensing with mineralogical- and geochemical analysis, and diffuse surface degassing measurements, we can infer the detailed anatomy of degassing and alteration systems at the surface, highlight active degassing domains versus areas of surface sealing, and determine their importance for the system based on their contribution to the total thermal energy release.

## 2 Study area

Vulcano is the southernmost of a group of 7 volcanic islands forming the Aeolian Archipelago north of Sicily. They are located within the Aeolian Tindary Letojanni Fault System (ATLFS), an NNW-SSE striking local deformation belt connecting the central Aeolian Islands with the eastern section of Sicily (Barreca et al., 2014; Cultrera et al., 2017). The ATLFS is the interface between two larger tectonically active compartments, an extensive one in the northeast and a contractional one in the west (Cultrea et al., 2017). Frequent seismic activity and right lateral extensional displacements indicate ongoing tectonic activity (Billi et al., 2006) and the active shaping of the islands.

Vulcano is made up of volcanic edifices of which the northern section of the islands is the most recently active. The oldest volcanic activity at Vulcano is reported for 130 ka (De Astis et al., 2013). Six main stages of volcanic activity have been identified (De Astis et al.1997), of which the geologically younger active parts, the La Fossa Cone and Vulcanello, have been active during historical times < 8 ka, showing mainly vulcanian and strombolian activity (De Astis et al., 2013). The last eruptive period of the La Fossa Cone from 1888-1890 was characterized by strong phreatic eruptions and witnessed and documented by Guiseppe Mercalli who later coined the term Vulcanian eruptions (Clarke et al., 2015).

Vulcano since then has been in a quiescent period and volcanic activity mainly expressed in degassing. Gasses are provided from a magmatic-hydrothermal system fed by a shallow magmatic reservoir beneath La Fossa volcano. The hydrothermal system is likely to have been partitioned into a hypersaline brine and a vapor phase (Henley and McNabb, 1978). The denser brine phase is confined at depth and contributes to the formation of metasomatic facies observed in deep-seated xenoliths (Adrian et al., 2007). The vapor phase, enriched with $SO_2$, $H_2S$, $HCl$, and $HF$, ascends to the surface and partly emerges directly from the high-temperature fumarolic field (Bolognesi and D'Amore, 1993; Chiodini et al., 2000; Capasso et al., 1997).

Volcanic degassing is present throughout the entire central and northern part of the island concentrating in degassing clusters at Baja Di Levante, within Vulcano Porto, and in clusters along the base and summit of La Fossa Cone (Chiodini et al., 1996;

Carapezza et al., 2011; Diliberto et al., 2021; Inguaggiato et al., 2022 and many others) where frequently higher fluxes of $CO_2$ are observed. The most prominent degassing sites are the high-temperature fumaroles at the summit of Vulcano that occur in several clusters on the outer rims of La Fossa cone and are most prominent in the high-temperature fumarole field (Figure 1). Degassing at the summit of La Fossa is persistent but subject to fluctuations. Gasses of the high-temperature fumaroles (HTF) emerge with temperatures > 300 °C, but temperatures have been exceeded during previous volcanic crises (Harris et al., 2012; Diliberto, 2017). Temperatures of up to 690 °C were reported in May 1993 by Chiodini et al. (1995).

Periods of unrest were accompanied by increasing fumarole temperatures (Harris et al., 2012; Diliberto, 2013; Madonia et al., 2013; Diliberto, 2017), increasing soil and groundwater temperatures (Capasso et al., 2014), changing gas compositions (Paonita et al., 2013), changes in gas flux (Inguaggiat et al., 2022), or a spatial growth of the fumarole field (Bukomirovic et al., 1997). The most recent crisis occurred in 2021 and led to increased thermal radiation (Coppola et al., 2022), deformation (INGV Bulletin reports), and localized structural changes like the formation of new major fumarole complexes. The rapid dynamics during volcanic crises and potentially negative effects of alteration on permeabilities, and therewith the potential to drain gasses from the surface, highlight the importance of a better understanding of the structure and state of degassing systems. Early studies about the structural setup of the Grand Cratere fumarole field of the La Fossa Cone were provided by Bukumirovic et al. (1997) and later modified (Madonia et al., 2016; Harris et al., 2009). Fulignati et al. (1999) analyzed alteration facies at Vulcano and constrained the central crater region to be a large silicic alteration complex characterized by the presence of chalcedony and amorphous silica. Outwardly to the central silicic alteration zone, advanced argillic (alunite ± gypsum) alteration develops, probably originating from the progressive neutralization of the acid fluids by weathering and dilution by meteoric waters (Fulignati et al., 1998). Müller et al. (2021) previously showed that degassing and alteration can be traced from remote sensing data far beyond the extent of the high-temperature fumarole locations. Based on surface color variability due to degassing and alteration processes, they showed evidence for a more complex setup with alteration gradients within the silicic alteration complex and important structural units that will be complemented here. Examples of degassing and alteration-related surface color variability are shown in Figure 2.

**3 Data and methods**

To analyze the degassing and alteration structure at Vulcano, we used a combination of UAS-derived remote sensing data (optical and thermal infrared imagery), image analysis, and field-based ground-truthing by mineralogical and geochemical analysis of rock samples and surface degassing measurements. A simplified sketch of the workflow is shown in Figure 3.

1) An anomaly detection (Chapter 3.2) based on UAS-derived data, employing image analysis techniques like Principal Component Analysis (PCA), and spectral and thermal classification (similar to Müller et al., 2021) provides the detailed optical and thermal anomaly pattern. Anomalies can be revealed based on slight color changes in the volcanic surface that occur due to degassing and hydrothermal alteration processes, or increased surface temperatures.

2) To verify observed anomalies, we carried out ground-truthing by mineralogical (XRD - X-ray diffraction) and geochemical
(XRF - X-ray fluorescence) lab analyses of representative rock samples. Further, we performed surface degassing
measurements to image the present-day degassing pattern and compared it to the observed anomaly pattern. Combining this
information we can infer a detailed anatomy of the degassing and alteration structure at the surface and define and parameterize
major structural units.
3) A temporal infrared monitoring carried out from 2018 to 2022, covering the volcanic crisis 2021 at Vulcano, allows us to
monitor the thermal evolution and response of the identified units to an event of increased gas flow with further implications
of critical processes like localized surface sealing. Details on the single analysis steps are provided below.
The remote sensing data used for this study and relevant processing steps are published in a Zenodo data repository
https://doi.org/10.5281/zenodo.12586672.

## 3.1. Acquisition and processing of UAS-based optical and thermal infrared data

The data acquisition was performed using a DJI Phantom 4 Pro quadcopter, equipped with a gimbal-stabilized 20 MP camera
with a real shutter system, recording up to 0.5 HZ. Optical overflights were performed in the daytime at an altitude of 150 m
above the fumarole field, ensuring a minimum overlap of 90% for later photogrammetric processing. Thermal infrared image
data was acquired by a Flir Tau 2 radiometric thermal infrared camera system attached to the DJI Phantom 4 Pro. The FLIR
Tau 2 measures in the spectral range of thermal infrared between 7.5 and 13 µm, has a resolution of 640 x 512 pixels, and is a
fully radiometric sensor system. The infrared image data is recorded at 8 Hz by a Teax Thermal Capture 2 data logger. The
camera was attached to the copter with a standard camera bracket on a self-made carrier frame and is powered by an external
11.1 V lithium-polymer battery, supplying voltage to the camera system (transformed down to 5.2 V in) and to an external
GPS antenna (> 8 V required) which provides coordinates for each infrared image. Infrared overflights were performed in the
early morning hours, before the sun illuminates the crater area, to avoid disturbances of irregular surface heating due to solar
radiation exposure (Stevenson and Varley, 2008). In this way, we ensure to map the thermal signal from the hydrothermal
system exclusively.
All image data were processed using the Structure from Motion (SfM) approach in Agisoft Photoscan (Version 1.5.2.7838).
The image data were inspected and images were preselected ensuring an overlap of 90%. Images of poor quality or out of
focus were excluded and only images of a constant flight altitude were used for the processing. This is particularly important
for the processing of infrared data, as varying altitudes might alter the radiation information due to changing pixel size to vent
ratios. The infrared data was pre-inspected in Thermoviewer 3.0 and exported in a 16-bit tiff format in grayscale. We followed
the typical workflow of sparse point cloud-, dense point cloud- and mesh generation, aiming to obtain a 3-dimensional model
and eventually orthomosaic, digital elevation model (DEM), and infrared mosaic. The original images and processing results
are roughly georeferenced, but their geolocation was optimized by manual co-registration using the ArcGIS georeferencing
toolbox. An overview of the acquired and processed data sets can be found in Table 1.

## 3.2 Anomaly detection - Principal Component Analysis (PCA) and spectral classification for alteration mapping


The alteration mapping was performed on an orthomosaic data set acquired in 2019 that due to poor fumarole activity provides
an almost distortion-free image of the central crater region. The alteration structure was revealed, similar to Müller et al.
(2021), by applying a Principal Component Analysis and image classification allowing further constraints on the zonation of
the fumarole area and expanding the interpretation by geochemical and mineralogical analyses of rock samples for ground
truthing. PCA is a statistical tool that was invented by Pearson (1901), further developed and widely applied in remote sensing
or image analysis (e.g. Loughlin, 1991; Fauvel et al., 2009; Alexandris et al., 2017). It can detect and highlight optical
anomalies within an RGB data set by transforming the data values of the initial RGB channels onto their perpendicular axes
of the highest data variance (e.g. Abdi and Williams, 2010). This can be achieved in several ways. We used the PCA
implemented in the ArcGIS image analysis toolbox (see ArcGIS online documentation for Principal Component Analysis),
performing the following workflow. In the first step, an ellipse including all data points is calculated for each dimension
(RGB). The main axes of these ellipses represent the Eigenvectors (direction of highest variance), and will be used as a new
coordinate system for the data transformation. By transforming all data points onto this new coordinate system, we obtain
Principal Components (PC) which are variance representations of the initial RGB bands and can be used to detect and highlight
optical anomalies like color changes due to alteration processes (Müller et al., 2021, Darmawan et al., 2022). PCA further
promotes a decorrelation of the initial RGB bands, a dimensionality reduction, and associated better data separability so that
color variations, before expressed by changes in the three RGB bands (3-dimensional problem), can now be accessed in single
bands, the single Principal Components (PC). While Principal Component 1 (PC) resembles 91% of the initial data variance,
it mainly shows brightness changes within the image. PC 2 and 3 contain 7% and 2% of the data variance, resemble color
changes, and are suitable to resolve optical anomalies related to hydrothermal alteration.
In our data, hydrothermally altered areas were defined based on the PC3, with pixel values > 85 representing hydrothermal
alteration. We used this as a mask to crop the respective pixel locations from the original orthomosaic (RGB), resulting in a
16 Mio pixel alteration raster subset (RGB). This alteration raster subset allows for a more sensitive image analysis due to the
reduced spectral range with respect to the original orthomosaic. Another iteration of PCA, now applied to the extracted
alteration raster subset adjusts to the new reduced spectral range, as we are excluding all redundant data e.g. unaltered surface,
and provide a variance representation of the altered surface exclusively. We classified the result in an unsupervised
classification (implemented in ArcGIS, using a combination of Iso Cluster and Maximum Likelihood classification) with 32
classes. We decided on unsupervised classification as this is a more data-explorative way of exploring the pixel information,
rather than classifying based on a spectral range constrained by training areas defined on pre-assumptions in a supervised
classification. The 32 classes are chosen to obtain a high class resolution, as this is the highest number of classes possible in
the unsupervised classification tool. By combining these classes in a way that they resemble larger optical spatial units, we
eventually defined 3 Types of alteration surface (Types 1 - 3) and the unaltered surface (Type 4) and further analyzed their
spectral characteristics and spatial distribution. Boxplots of the distribution of RGB values in the 32 classes and the spectral
range of Type 1 - 4 surfaces are shown in Appendix A. The optical structure of the fumarole field and alteration zone is similar
to the thermal structure and will be discussed in Chapter 4.2.

**3.3 Infrared analysis - thermal structure and time series analysis**

The SfM-derived infrared mosaic represents the thermal radiation in a 16-bit tiff format, resembling values between 0 and
65536. To obtain a temperature map from the IR mosaic we calculated the apparent pixel temperatures Tp by
$$Tp \text{ (in K)} = grayvalue * 0.04 \quad (1)$$
where Tp is the apparent pixel Temperature in K, the gray value is the radiation value of the original infrared mosaic and 0.04
is the scaling factor (radiometric resolution). The temperature map was used to define the thermal structure. We observed
several distinct thermal spatial units with temperatures significantly above the background temperature, that can be
distinguished in high-temperature fumaroles (HTF in the following) and areas of rather diffuse thermal surface heating (Figure
4 B/D). To constrain these units spatially for further comparison, we had to approximate spatial boundaries what was done
after comparison to our optical data and based on knowledge of previous observations by defining the temperature thresholds
of T = 22 – 40 °C for the diffuse heated areas and T > 40 °C for HTF. The 40 °C threshold resembles well the known locations
and extent of HTF in the upper and lower fumarole field. To compare the thermal emissions of detected structural units, we
calculate the radiant exitance values by applying the modified Boltzmann law.
$$Q = e*b*A*(Tp^4 - To^4) \quad (2)$$
The emissivity (e) was assumed to be 0.95 (often used as an assumption for volcanic surfaces), the Boltzmann constant (b) is
$5.670737 \times 10^{-8}$ $Wm^{-2}K^4$, the area of a pixel (A) is 0.024 m². *Tp* is the pixel temperature and *To* is the average background
mean temperature, calculated based on 9 reference areas that are anomaly-free. To compare identified units quantitatively, we
summarized the radiation per pixel for the respective units a - g to a cumulative thermal radiation (Rcum). The flight altitude
of 150 m (above the fumarole field ) in combination with the low resolution of infrared sensors results in a pixel resolution of
0.38 x 0.38 m.
Note that remotely sensed Infrared data always represents apparent temperatures that might differ from the real object
temperature due to the radiation properties of the measured object itself (emissivity), the distance of the sensor to the measured
object, the pixel-to-object size ratio, but also due to atmospheric or hydro-meteorological effects (Ball and Pinkerton, 2006)
influencing the detected radiation values. Therefore apparent temperatures typically are lower than in situ vent temperatures.
Real fumarole vent temperatures can reach more than 300 °C (Diliberto, 2013) while temperatures in our infrared mosaic range
to max. 163 °C only. With this data set, we do not aim to provide precise fumarole temperatures but to analyze the thermal
structure of the fumarole field and the broader field of activity.

**3.4 Ground-truthing by Mineralogical and Geochemical Analysis**
**3.4.1 Rock sampling**
Rock samples were collected at predefined representative locations aiming to include all alteration end members, during field
campaigns in 2019 and 2022. We sampled along 3 transects following the postulated hydrothermal alteration gradients and
crosscutting major alteration units, of which transect A is located on the lower fumarole field, transect B along the upper crater
rim, and transect C in the eastern fumarole field (locations for samples in Figure 7). Samples were in the size of ~2000 cm³
(hand-sized) retaining the undisturbed surface crust, but also subsurface material to a depth of ~10 cm. The samples were
mechanically crushed, ground to 63 µm, and split for XRD and XRF analysis, respectively. In total 21 samples were collected
of which 9 were prepared for the XRD and XRF analysis and 12 for XRF analysis exclusively.
**3.4.2 X-ray diffraction (XRD)**
Between 1 and 3 mg of whole rock powder was used to determine the mineral composition of each sample through powder X-
ray diffraction (pXRD). The analysis was conducted using a PANalytical X'pert diffractometer equipped with an X'Celerator
silicon-strip detector at the Department of Geoscience, Swedish Museum of Natural History, Stockholm. The instrument was
operated at 45 kV and 40 mA using Cu-Kα radiation ($\lambda = 1.5406$ Å). Samples were analyzed between 5° and 70° (2θ) for 20
min in step sizes of 0.017° in continuous scanning mode while rotating the sample. Data were collected with "divergent slit
mode" and converted to "fixed slit mode" for Rietveld refinement. The collected data show several peaks of X-ray diffraction
intensity which represent the characteristic of crystalline minerals, the proportions of mineral phases were then refined using
the Rietveld refinement method in the High Score Plus 4.6e software. The XRD analytical procedure was performed twice for
each sample to ensure optimal quality control. Some samples contained contents of amorphous material of more than 50%.
Those will be marked with a * in the following but we will consider the mineral composition normalized to 100% non-
amorphous material.
**3.4.3 X-ray fluorescence (XRF)**
Bulk chemical composition was determined by X-ray fluorescence analysis (XRF) at the ElMiE Lab at the German Center for
Geosciences (GFZ). Main and trace elements were measured on fused beads with an AXIOS spectrometer (Malvern
Panalytical, UK). Loss of ignition (LOI) was determined by analysis of $H_2O/CO_2$ using an Eltra element analyzer.
Reproducibility was determined on three certified reference materials (CRM) and is within the analytical precision, which is
better than 2% for main elements and better than 10% for trace elements.
**3.4.4 Surface degassing measurements ($CO_2$, $SO_2$, $H_2S$)**
The surface degassing was measured at ~200 points within the northern part of the La Fossa cone (Figure 6) in September
2021 and November 2022 using a simplified Multigas accumulation chamber approach (Appendix B). The measurement unit,
a Dräger X-am 8000 handheld Multigas device, was equipped with 6 sensors measuring $CO_2$, $CH_4$, $SO_2$, $H_2S$, $H_2$, and $O2$
simultaneously of which $CO_2$, $SO_2$, and $H_2S$ are considered here. The simplified accumulation chamber approach was an
adaption as a consequence of uncertainties encountered in previous Multigas measurement campaigns. Due to different sensor
reaction times for ascending and especially descending gas concentrations, the comparison of direct sensor readings might
lead to odd gas ratios with an artificial shift towards magmatic components. For that reason, we use the slope of the ascending
gas concentration within a defined volume as quantification for a relative surface flow. More detailed information about the
gas measurement approach is provided in the supplementary materials. Note that the aim of the gas measurements was not to
provide accurate flux estimates but to highlight the spatial variability of the gas flow of certain gas species from the surface.
**4 Results**
**4.1 Thermal- and optical anomaly patterns reveal the degassing and hydrothermal alteration structures**
Degassing and hydrothermal alteration at La Fossa as seen in drone imagery can be traced by mainly two effects.
1) The transition from unaltered to hydrothermally altered surface can be traced by a general color shift in the drone images
from reddish to grayish (Figures 2A and 4A). This allowed us to constrain a distinct ~70,000 m² sized area surrounding the
fumarole field in a circumferential manner. This area is hereafter referred to as the Alteration Zone (ALTZ in Figure 4A/B),
and represents the maximum extent of at the surface observable alteration effects that can be associated with the fumarole
field. It includes effects ranging from weak surface alteration to strong alteration with intense surface bleaching and
remineralization, to complete destruction of the host material. The extent of the ALTZ exceeds the area covered by the high-
temperature fumarole (HTF) site by ~60 times (Figure 4A/B), indicating the widespread influence of diffuse degassing and
alteration processes.
2) Within the ALTZ we observe a segmentation characterized by brightness and color variability expressed in different shades
of gray (Figure 4A), the second optical effect, indicating local alteration gradients. Analyzing the ALTZ for this spectral
variability by PCA and image classification we can constrain pixels of low-, increased-, or intense surface bleaching and
alteration (light blue, dark blue, and red pixels in Figure 4B) and define an alteration index represented by 4 surface Types (1
- 4), of which Type 1 is the most altered and Type 4 the least altered / unaltered surface.

Type 1 surfaces are bright grayish intensely bleached surfaces or sulfuric deposits and represent the strongest alteration end members that we can detect optically from our data. Type 1 mainly resembles the fumarole sites and surrounding areas (Figure 4 A/B) but also larger isolated regions that can not be associated with major vent systems. With increasing distance to the degassing centers, we observe a shift towards darker grey (Type 2) and brownish (Type 3) surface colors. Type 2 is characterized by a gray but comparatively less bright coloring. It typically occurs at the boundaries between Type 1 and Type 3 surface and largely surrounds Type 1 areas, but it also forms several isolated clusters typically embedded in Type 3 areas (units b,d,g in Figure 4). Type 3 is generally darker and more reddish in color, similar to the unaltered parts of the crater surface, but can be well distinguished from the unaltered surfaces by PCA. It makes up ~50% of the ALTZ and dominates in the central northern and the southeastern parts. Type 4 is a reddish, apparently oxidized surface that dominates the La Fossa cone surrounding the ALTZ.

The surfaces within the ALTZ are generally mixed and composed of more than one type. The ALTZ is characterized by a generally high density of Type 3 pixels, with locally high densities of Type 1 and Type 2 pixels, which then become the dominant surface Type and form larger spatial units, indicating locally higher alteration gradients or larger structural units (units a - g in Figure 4 and details in Appendix C). The largest of these units cover several thousand square meters each.

The thermally active surface (Figure 4C) can be divided into high-temperature fumaroles (HTF in Figure 4D) and diffuse thermally active surface (green pixels in Figure 4D). HTFs are the visible part of the activity that can be constrained by the naked eye in the field, while the diffuse thermally active surface is largely imperceptible. The thermally active surface largely mirrors the alteration pattern observed in the optical data. An analysis of the temperatures obtained at all pixels of Type 1 to 4 surface shows a general decrease of mean pixel temperatures from Type 1 to Type 4 surface by an average of 2 degrees (Figure 5). In particular, areas dominated by Type 1 and 2 surfaces reflect the thermal structure well while areas of Type 3 dominance largely coincide with low-temperature surfaces (Figure 4 B/D). An additional Spearman correlation test, applied to the classified surface (32 classes unsupervised, for comparison see Appendix A) and the thermal data (in °C) shows a moderate positive correlation between optical and thermal anomalies (Appendix D). This shows that the detected optical anomalies are meaningful and that degassing and alteration variability occurs even at local scales and can be traced in our close-range drone remote sensing data.

The spatial coincidence of both optical and thermal anomalies highlights the relationship between variations in the surface coloration, caused by alteration processes, and the ongoing influence of diffuse gas flow. A general coincidence of increasing brightness (simultaneously increasing the RGB values) with increasing surface temperature of an area can be constrained (Figure 5).

**4.2 Anomaly structure identified from optical and thermal data**

The optical and thermal anomalies form distinct spatial units of alteration and elevated surface temperature (units a - g in Figure 4D), which now allow us to infer the following surface structure of the fumarole field and its wider field of activity.

The centers of degassing activity are high-temperature fumaroles (HTF). These are the "visible" parts of the activity that can be perceived in the field (red pixels in Figure 4D). We spatially constrain the HTFs based on apparent temperature values with $T > 40$ °C in our 150 m overflight data. Using this as a threshold we find that the HTFs cover an area of 1223 m², and locate exclusively in the Type 1 surface. However, HTFs represent only a fraction of the active surface.

The total extent of the surface that has to be considered active is much larger. The surface with elevated temperature covers ~30,000 m² (green pixels in Figure 4D, $T > 22$ °C or 5 °C above the background), exceeding the area covered by HTF by a factor of 25. The surface that is considered hydrothermally altered (ALTZ ~70,000 m²) exceeds the area covered by HTF by a factor of ~60, highlighting the widespread influence of diffuse degassing and alteration processes.

Besides the HTF we have constrained larger units of elevated surface bleaching and surface temperatures that can be considered structurally important and centers of diffuse degassing activity.

Units a and b are diffuse features of increased surface bleaching (Type 1 and 2) and surface temperature, embedded in the Type 3 surface and surrounding the eastern fumarole field in the form of an aureole shape. Neither can be associated with major vents. The observed maximum surface temperature for unit a is 43.7 °C (0.38 m resolution) and the average temperature is 25 °C, ~8 °C above the background. It is located at a distance of 25 to 50 m downslope from the eastern rim fumarole complexes, separated by a Low-Temperature Zone (LTZ). Unit b, the southern part of the aureole is a 120 m long and 20 - 35 m wide anomaly located subparallel on the inner side of the crater. It extends over ~2100 m² and has a maximum surface temperature of 46 °C and an average temperature of 26 °C (9 °C above the background). The temperature range and spatial extent of units a and b are comparable. In the field, both are difficult to identify as there is little or no evidence of degassing (Appendix E). Like unit a, unit b is also separated from the main fumarole vents by the LTZ. Its northern boundary corresponds exactly to the positions and curvature of the fumarole alignments at a relatively constant distance of 30 meters. In unit b, we observe a temperature gradient with higher temperatures at greater distances from the fumarole vents and an apparently more active center in the southeastern corner. Another thermal anomaly with a similar shape and orientation is located further south inside the crater.

Units d and f are similar aureole-like features in the western fumarole field, associated with fumarole complex F0. They circumferentially surround fumarole complex F0 at a distance of 5 to 15 m, also separated from the HTF by a Low-Temperature Zone (LTZ), but to a lesser extent than that observed for units a and b of the eastern fumarole field. The southwestern section of this aureole, unit d, appears as a larger heated complex with stronger surface bleaching (Type 1) and higher temperature (mean $T = 27$ °C), and a temperature gradient with higher temperatures further away from the major fumarole complex. The boundary to the Low-Temperature Zone is sharp with a sudden drop in temperature of $10 - 20$ °C and a strong associated color shift (Appendix C). The aureoles of F0 (d,f) and F11 (a,b) have in common that they are encircled by a network of polygonal net-shaped thermal anomalies in the far field.

Low-Temperature Zones (LTZs) dominate the central parts of the fumarole field. The LTZ have only slightly elevated temperatures relative to the background ($18 - 21$ °C or $1 - 4$ °C above background) and can be optically constrained by a darker Type 3 surface coloration. From field observations, we have concluded that these LTZ are strong, apparently sealed surface complexes. Therefore LTZ might indicate largely sealed sections of the fumarole field which inhibit gas flow at the surface. The 3 central LTZ 1 - 3 (Figure 4D) cover an area of ~12,000 m².

Unit c is a broad complex of highly altered material (Type 1) and significantly high surface temperatures. It is potentially the most altered unit in the central crater region. It covers an area of ~8000 m² and the maximum and average apparent temperatures observed are 87 °C and 29 °C. It is associated with the HTF FA and F58. Considering the thermal structure of unit c, it is a heterogeneous unit composed of a network of higher temperature anomalies embedded in lower but, with respect to the background, significantly increased tempered surface. This area is associated with the northwestern crater unit, which is the most recent explosion crater.

Unit d is an area of diffuse activity associated with the inner crater part of fumarole complex F0, showing a significant shift in surface colorization and temperature, some 20 m away from the fumarole complex. The boundary is distinct and visible to the naked eye (Type 3 to Type 1). The temperature shift is on the order > 20 °C.

Unit e is a large branching thermal and optical anomaly of the upper fumarole field. It can be constrained by its gray coloration embedded in the reddish unaltered surface and also by its increased surface temperature. It is a 120 – 150 m long branch-shaped network of anomalies on the inner crater wall. The central feature is oriented E-W and located ~20 m south and below the helicopter platform and the crater rim. We constrained its size to ~2500 m² (only the western branch, without intersection to unit d) and the recorded maximum and average apparent temperatures are 45.0 °C and 25.9 °C respectively. Some smaller clusters of localized degassing, alteration, and increased surface temperature, visible at the surface by its bright coloration, are observed in the northern section of the fumarole field (unit g) towards La Forgia.

## 4.3 Ground truthing - verification of observed anomalies

We have carried out mineralogical (X-ray diffraction) and geochemical (X-ray fluorescence) analyses of bulk rock samples collected at representative locations and surface degassing measurements. The aim is to verify the observed anomaly pattern of alteration gradients and distinct active units and to investigate the relationship between the optical and thermal anomaly pattern and modern degassing and hydrothermal alteration processes. In this way, we provide ground truthing and demonstrate that the anomalies observed are significant.

### 4.3.1 Present-day surface degassing pattern

The measurements of diffuse degassing from the surface allow us to compare the present-day surface degassing pattern to the observed optical and thermal anomalies (Figure 6 A/B). We performed surface degassing measurements of $CO_2$, $H_2S$, and $SO_2$ simultaneously in the diffuse degassing regime at 200 measurement points (~100 points within and outside the ALTZ) throughout the whole northern crater section (details of gas measurements in Appendix B).

The observed relative flux values for $CO_2$ range from 0 to ~9000 ppmv/s with an average of ~900 ppmv/s. They are considerably higher (x*10³) than the $SO_2$ and $H_2S$ flux at the respective locations. For both, $SO_2$ and $H_2S$, a maximum gas flux of < 10 ppmv/s was measured and the average is < 0.5 ppmv/s.

Looking at the spatial distribution of the measured gasses we observed generally higher gas levels within the alteration zone
ALTZ and at the ALTZ boundary, for each of the measured gasses (Figure 6 C/D). The average $CO_2$ flux is 660 ppmv/s outside
the ALTZ and 923 ppmv/s within the ALTZ. Thus, the averaged $CO_2$ flux inside the ALTZ is about 1.4 times higher than
outside but is particularly high in some of the constrained units a - g. However, the $CO_2$ flux has a wide spatial distribution
and high flux values of above 2000 ppmv/s can also be observed outside the ALTZ and at a distance to the ALTZ boundary
(Figure 6 A/C).
$SO_2$ and $H_2S$ in contrast appear spatially stronger confined, and significant flux values can be exclusively observed within the
ALTZ (Figure 6 B/D). Values for $SO_2$ and $H_2S$ inside the ALTZ exceed the outside-ALTZ values by 13 and 15 times. This
higher diffuse flux, although at average low concentrations, might promote a surficial process of chemical weathering and
surface bleaching, potentially causing the observed color shift from a reddish-oxidized surface toward gray and will be
discussed further based on analyses of the geochemical composition of rock samples in Chapter 5.2.
Comparing the surface degassing to the observed optical and thermal anomaly pattern (Figure 6 E-G), we see that high values
were observed especially in units a or b on the eastern side of the fumarole field, coincident with increased alteration (Type 1
and 2) and thermally active surface, followed by other constrained units c - g. However, the strongly bleached and apparently
highly altered unit c shows, other than expected, rather small gas fluxes, although its surface temperature is significantly
increased with respect to other identified units. This might indicate reduced surface permeability and surface sealing processes
and will be discussed in Chapter 5.3.
While $SO_2$ or $H_2S$ flux values for Type 1 and 2 surfaces are increased, only low fluxes were constrained for the Type 3 surface
and no flux for the unaltered surface (un in Figure 6 E-G). Note that the central sections of the fumarole field were not sampled
due to the close vicinity to HTF and expected high flux values. The data shown here is only representative for the diffuse
degassing domain.

### 4.3.2 Mineralogical composition of the alteration gradients

XRD Analysis was performed along two transects A and B, and XRF analysis was performed on samples taken along three
transects A - C (Figure 7), crossing postulated alteration gradients.
Transect A crosses from the unaltered surface over Type 3 into the Type 1 surface (T3 - T1 boundary in Figure 7) of the highly
altered unit c. Transect B is oriented along the HTF on the crater rim in an east-west orientation from Type 1 surface into the
LTZ (Type 3). Transect C crosses the eastern fumarole field from the unaltered surface, through the Type 1 and 2 surfaces in
unit b, the LTZ (Type 3) on both sides of the HTF, to Type 1 and 2 surfaces, and eventually the unaltered material outside the
ALTZ. This transect represents the variability in the rather diffuse degassing regime as no samples close to the HTF were used
for the analysis.

Results of all samples support local alteration gradients within the ALTZ and show significant changes in the mineralogical and geochemical compositions (Table of XRD results in Appendix F). The dominant mineral phases observed in samples of transects A and B are sanidine, cristobalite, and elemental sulfur (Figure 7). Additionally, most samples contain amorphous material, representing glassy phases typical for volcanic sequences. For comparability, mineralogical concentrations refer to the crystalline phase, while amorphous contents are stated with respect to the total. Note, however, that bulk rock geochemistry refers to both phases and cannot analytically distinguish between amorphous and crystalline.

Considering compositional changes along transect A, we observe a high proportion of sanidine feldspar and lesser cristobalite in the relatively unaltered samples (Type 4). With an increasing degree of alteration, we observe a general loss of cristobalite and sanidine while sulfur contents increase (Figure 7). Samples from the unaltered reddish Type 4 surface (sample A1) outside the ALTZ and Type 3 surface (samples A 2/3) inside the ALTZ are similar in composition and show high sanidine and cristobalite contents of 86 - 87% and 13 - 14% in the crystalline part, respectively, yet low to no sulfur contents. These samples were taken in areas of no or only slightly increased surface temperatures of < 22 °C (i.e. < 5 °C above the background). Samples A 4 - 6 are taken in unit c, a complex of high alteration and increased mean surface temperatures of 29 °C (> 10 °C above background). In this strongly altered unit, cristobalite is absent, along with a decrease in sanidine to 60 - 70% relative to the least altered samples and an increase in sulfur contents of up to 20 - 40% in the crystalline portion of the rock sample. However, the amorphous components constitute a high proportion of these sample(s), showing ca. 50% in samples A6 and B1.

Samples taken on the upper rim along transect B in the high-temperature fumarole regime contain total sulfur contents of 50 to 100%, while cristobalite is absent in these samples. Sample B3, a piece of grayish crust is taken from LTZ 3 (~21 °C, 4 °C above background) in between the high-temperature fumaroles F0 and F5 and contains 100% sulfur, highlighting the precipitation and sealing potential of degassing activity at the surface.

Comparing the changes of surface coloration with changes in the mineralogical composition we can constrain no significant effect at the ALTZ boundary, i.e., the transition from unaltered to altered surface (A1 – A 2/3), although the optical effect is major. However, significant compositional changes, e.g. the complete loss of cristobalite and increasing sulfur content are observed at the Type 3 – Type 1 boundary (T3 - T1, blue mark in Fig. 7), coincident with the shift from Type 3 to Type 1 surface into unit c.

### 4.3.3 Bulk geochemical composition of the alteration gradients

For samples without amorphous fraction, bulk geochemical composition correlates reasonably well with mineralogy determined by XRD, assuming ideal stoichiometry. The difference between theoretical bulk composition and true composition is within 10% of the respective element, which we consider a good estimate given sample heterogeneity. Only for sample A5, the high Mn content remains unmatched by XRD analysis. Subtracting the theoretical bulk composition of the crystalline fraction from the true bulk composition, we can thus estimate the chemical composition of the amorphous fraction. The amorphous fraction is similar to the crystalline counterpart mainly composed of $SiO_2$ and some minor (< 5 wt.%) phases, as

well as elevated Mn contents. The high Mn contents were only observed in samples with medium alteration and elevated temperatures, both in samples with and without a significant contribution portion of amorphous material. It is thus likely that Mn is contained in the crystalline phase, yet could not be detected due to the high $SiO_2$ signals derived from sanidine and amorphous material.

The bulk geochemical composition (Figure 7 and data table in Appendix G) agrees with the mineralogical composition. All samples are high in $SiO_2$ content and, therefore, can be considered to belong to the large silicic-alteration complex earlier described by Fulignati et al. (1999). The samples show a slight variability of $SiO_2$ between 67 - 82 wt.% and plot on the rhyolite field within the TAS diagram (Middlemost, 1994; not shown here). The amorphous component, typical for rhyolite, consists of mainly $SiO_2$, with minor amounts of Fe and Al, based on the difference between the theoretical and actual geochemical composition calculated from stoichiometric mineralogy. Three samples also have significant MnO, possibly caused by hydrothermal leaching and precipitation as amorphous crusts. However, the variability of MnO will not be detailed further in this study.

Dominant in transect A is the loss of $Al_2O_3$ from the unaltered Type 4 surface ( > 10 wt.%) outside ALTZ to the Type 1 surface of the highly altered unit c (< 0.4 wt.%). Similarly, $Fe_2O_3$ is decreasing from an average of 1.6 to 0.3 wt.%. The loss of $Al_2O_3$ and $Fe_2O_3$ is likely related to the alteration of sanidine and the elution of iron- and aluminum-sulfates formed due to the contact with sulfuric gas. The most significant changes occur, similar as observed in the mineralogy, not at the transition from unaltered to altered (ALTZ boundary) but at the T3 - T1 boundary (blue line in Figure 7) at the transition from Type 3 to Type 1 surface. Transect C crosses from the unaltered surface through unit a, the northern LTZ, the southern LTZ, unit b, Type 3 surface, and eventually the unaltered surface. Compositional changes from unaltered (Type 4) to altered (Type 1 and 2) surface of units a and b, here, are minor with relatively stable values for $Al_2O_3$ (6 - 12%), $Fe_2O_3$ (1 - 3%), $TiO_2$ (< 0.5%) and Mn (~0%). At the transition from active units a and b (Type 1 and 2 surface) to the LTZ (Type 3), we observe a significant increase of sulfur content from < 2% to 12 - 40%. However, this increasing sulfur content here is not coincident with the systematically brighter surface color observed for other altered units. LTZ show the same rather dark surface observed for Type 3 surface elsewhere, which is a discrepancy to the effects observed in the western fumarole field and indicates that LTZ have to be considered subject to different surficial processes. This will be discussed in Chapter 5.3.

**5 Discussion**

Various studies have previously explored the geochemistry (e.g. Fulignati et al., 1999), petrology (e.g. De Astis et al., 1997), geophysics (e.g. Revil et al., 2008) and remote sensing signal (e.g. Mannini et al., 2019; Coppola et al., 2022) of the La Fossa crater, Vulcano island. In this study, we combine close-range remote sensing, image analysis, mineralogical and geochemical analyses of rock samples, and the study of the present-day surface degassing of the La Fossa crater and analyse the surface expression of hydrothermal activity. Through this combination of ultra high resolution (< 10 cm) drone image analysis with

mineralogical/geochemical analysis, we are able to provide a holistic picture of the surface degassing and hydrothermal alteration pattern, highlighting an aureole-like organisation of the alteration field that distinguishes distinct units that grade from inner high-temperature fumaroles to sealed surfaces (LTZ) and to diffuse degassing areas at a greater distance. An area of approximately 70,000 m², which we have termed the Alteration Zone (ALTZ), outlines the maximum extent of observable alteration effects and highlights that degassing and alteration can be traced well beyond the central high-temperature fumarole activity sites (HTF). The ALTZ is similar to the diffuse flux zone previously identified by Mannini et al. (2019), and is in fact ~60 times larger than the area covered by the high-temperature fumarole domain. However, from the optical data, we observe further variability within this zone expressed in active units a - g, which also coincide with the diffuse thermal activity and cover 25 times the extent of HTF. Here we can further detail the surface structures and activity patterns identified in previous works, (e.g. Harris et al., 2009; Mannini et al., 2019) which divide the active region into a vent flux zone and a diffuse flux zone. In particular, we can further detail the vent flux zone by outlining high-temperature fumarole locations based on high-temperature pixels. We further show that thermal radiation and gas flux of the ALTZ or diffuse flux zone are not uniform but show strong local variability, with high fluxes in identified active units a - g and low fluxes in larger parts of the central fumarole field associated with LTZ (Low-Temperature Zones) or the unaltered regime.

Although the structural study is based solely on drone-derived imagery and thermal infrared data, our detailed observations of local activity and alteration gradients are supported by variations in mineralogical and bulk geochemical compositions of representative rock samples. Although we have performed a classification of the image data in this work, already visual observations can show a general shift from reddish to gray surface color, which coincides with areas of increased diffuse gas flow. Such color anomalies can therefore be used as guide in the field. A variability in surface brightness and gray hues coincides with alteration gradients and major active units, and the congruent optical and thermal anomaly pattern indicate the link between surface coloration and degassing-induced alteration processes. This relationship underlines the potential of the presented combination of methods as an efficient first-order site investigation tool for volcanic degassing and alteration systems that can be applied to volcanoes elsewhere.

### 5.1 Alteration Zone (ALTZ) controlled by sulfuric gasses and elution processes

The ALTZ is characterized by a surface color shift from reddish to gray and coincides with higher $SO_2/H_2S$ flux and appears to represent a zone of diffuse acid gas flow (Figure 4). All measurements with a significant flow of sulfuric gas species are from inside the ALTZ, while the flux of $CO_2$ was high well beyond the ALTZ. We, therefore, suggest the general color shift from reddish to gray to be related to a higher flux of sulfuric gasses, promoting chemical leaching of iron oxides via the reduction of the initially contained iron oxides to iron sulfates, which are strongly soluble in rain or in condensing water vapor and are thus prone to rapid elution. Iron oxide content in our analyzed samples ranges from 1.5% (sample A1) in the unaltered regime, to 1.1 - 1.4% for samples A 2/3, and to 0.3 for samples A 4 - 6 of the highly altered unit c. There is a gradual reduction

following the postulated alteration gradient, with the strongest changes along the T3 - T1 boundary (blue line in Figure 7). The 1.5% $Fe_2O_3$ for our rock sample of the unaltered regime is a rather low value and might be related to the fact that the sample was taken very close to the ALTZ boundary. It consists of > 50% amorphous material. Fulignati et al. (1999), who provide a broader sampling database estimated $Fe_2O_3$ contents of unaltered 1888 - 1890-eruptive products with 2.5 - 6.7%, which reduces to an average value of below 1% in the silicic-alteration regime (Fulignati et al., 1998; Fulignati et al., 1999; Boyce et al., 2007).

Further evidence for chemical leaching is found on the crater floor, where deposits form a colored layer resembling the color spectra widely observed on La Fossa, with bright reddish deposits close to the fumarole field resembling fluvial patterns. We believe that the optically anomalous gray surface at Vulcano can generally be used to infer areas of present higher acid gas flux or former discharge of acid gasses. Analyzing the broader area of the central crater region, we can infer multiple other areas where we observe similar changes of colorization that indicate similar argillic or strong silicic alteration effects at the surface. These are located on the southern inner crater, the outer crater rims, the 1988 landslide area (Madonia et al. 2019) and the northern flank towards Vulcano Porto. These zones of strong alteration are indicated in red in Figure 1B or Müller et al. (2021).

## 5.2 Alteration gradients on local scales

With average high $SiO_2$ contents of > 70%, the sampled areas correspond to the large silicic alteration complex suggested by Fulignati et al. (1999), Azzarini et al. (2001), Boyce et al. (2007), and others. In our study, we show evidence for strong local alteration gradients and structurally important units and spatially constrain them, thus we complement earlier studies.

Color shifts observed within the ALTZ associated with units a - g (brightness effects, hues of gray) are likely controlled by the degree of hydrothermal alteration, secondary mineral formation, and especially sulfur content in the respective surface samples. Coincident with characteristic changes in the surface coloration from Type 4 towards stronger bleached surfaces of Type 1, we observe a relative decrease of the initial mineral and element composition and a simultaneous increase in sulfur content for most of the obtained samples (Figure 7). While sulfur content in Type 1 surface ranges from 6 to 31%, for Type 2 it is already below 2%, and for the unaltered fraction, it is below 0.2%. We can, therefore, confirm a general link between alteration gradients, sulfur contents, and surface brightness (or surface types) in our remote sensing data (Figure 8).

An exception from this trend are sulfur contents of the Type 3 surface. Here we observe two distinct clusters (Figure 8), one with values below 0.5% and one with extraordinarily high sulfur contents of 12 to > 60%, both showing a similar surface coloration. All Type 3 samples with high sulfur contents are exclusively taken from LTZ. This strong discrepancy of sulfur content and surface coloration within the Low-Temperature Zones suggests next to alteration gradients also surficial or shallow processes of mineral deposition and formation of sulfur-rich encrustations that form sealed surfaces, especially in the near field of fumaroles. The low temperatures observed within LTZ and the limited surface degassing highlight the efficiency of such sealing processes. So far we can not distinguish LTZ from Type 3 surfaces in our optical data (Figure 8). A distinction,

however, would be beneficial as it would provide a method allowing for the precise spatial constraint of sealed surfaces from simple UAS-derived RGB imagery.

The intensity of optical and thermal effects and associated changes in mineralogical and chemical composition, and degassing are not always equally significant. Although the general shift from unaltered surface to altered surface (ALTZ boundary, shift reddish to gray) is a major criterion for the identification of degassing and alteration extent in our data, the associated changes in compositions are minor (Figure 9). The larger changes are observed within the ALTZ at the T3 - T1 boundary (Figure 7). Here we observe a sudden decrease in the initial mineral and bulk geochemical composition and an increasing sulfur content. We interpret the rather low changes at the transition from unaltered to altered at the ALTZ boundary to be related to rather weak or surficial alteration effects. The size of obtained samples was on the order of ~2000 cm², including the surficial part, but extends down to ca. 10 cm depth. This way, it was not possible to trace mineralogical or geochemical changes at the surface only. The samples obtained at the T3 - T1 boundary, on the other hand, show strong changes and reveal the general systematics of alteration effects, especially those samples taken in unit c, which might be considered one of the strongest alteration end members of the central crater region.

The identified surface patterns with respect to alteration gradients and structural units result from long-term evolution. However, some features may be subject to rapid changes. During the volcanic crisis in 2021, we observed, for instance, the formation of a new fumarole complex, which will locally change the composition, sulfur content, and surface type considered (cf. Figure 11), as is the case with sulfur deposits at the surface that can change quickly due to rainfall.

## 5.3 Heat budget - evidence for diffuse activity and surface sealing

Heat budgets on Vulcano have been studied earlier by e.g. ground-based surveys or satellite data (Chiodini et al., 2005; Harris et al., 2009), providing a range of estimates of thermal emissions (Mannini et al., 2019; Silvestri et al., 2019; Coppola et al., 2022). We compare our results to those using a remote sensing approach. Mannini et al. (2019) outlined a diffuse flux zone, which is comparable to the ALTZ defined by us. Further, they divided this zone into a diffuse flux zone and a vent flux zone and estimated the thermal radiation.

We outlined the surface structure of the degassing and alteration system (simplified in Figure 10 and detailed views of the different alteration and thermal units in Appendix C) based on the detection and classification of optical and thermal anomalies supported by additional mineralogical, geochemical, and degassing information, and spatially constrained high-temperature fumaroles (HTF), major diffuse active units a - g, and Low-Temperature Zones (LTZ). We quantified their importance for the degassing and alteration system by calculating their thermal energy release according to the Boltzmann Law (Eq2 and Figure 10) for both, anomalies with T > 22 °C, and identified units based on a spatial constraint, also including values < 22 °C. Therefore our identified structure differs from the vent flux and diffuse flux zone shown by Mannini et al. (2019).

High-temperature fumaroles in our study have average radiant exitance values of 82 W/m² but can only account for 28% of the total emitted thermal energy (calculations based on pixels with T > 22 °C and a corrected background of T = 16.71 °C).

Note that when considering pixels with T > 40 °C (as outlined by red patches in Figure 4D) the cumulative radiation of HTF would decrease to 0.2 MW only and the radiant exitance would increase to 242 W/m². The rest is released by the diffuse degassing part of the system. Although a direct comparison may be difficult, due to different outlines chosen for the respective flux zones, different sensor systems, and different background correction values, our cumulative radiation (Rcum in Figure 10) for the HTF is comparable to those of Mannini et al. (2019). These authors suggested a vent zone heat flux of 0.35-0.96 MW. Our estimate was ~1 MW, of which 0.5 MW are contributed by the high-temperature fumarole vents and 0.52 MW by unit c. This is also in accordance with the findings of Coppola et al. (2022), who estimate a similar vent zone flux from VIIRS imaging bands.

However, the contribution of the diffuse thermal regime approximated by us is lower compared to estimates in other studies (e.g Mannini et al., 2019), which place the contribution of the diffuse flux zone (comparable to our ALTZ) on the order of 90% of the total flux. This is likely related to the fact that we did not estimate the flux of the ALTZ in total, but of multiple larger anomaly units within the ALTZ that have apparent structural importance for the degassing system and contribute ~50% to the total flux. For future studies, a distinction into 3 different thermal regimes, a vent flux / high-temperature fumarole zone, diffuse active units (like a-g shown in this study), and a broader low-temperature anomaly field (diffuse flux zone/ ALTZ) may be recommended to better resolve close range thermal-infrared remote sensing information with data from satellite-based studies.

Regarding the heat budget estimated by us, the most dominant diffuse unit in terms of thermal energy release is unit c. It has the second-highest average radiant exitance of 76 W/m² and exceeds, with 29% of the total thermal radiation, the cumulative radiation of the HTF. Unit c is not only a highly altered complex with the strongest bleached surface and increased surface temperatures, there is also a discrepancy to the current degassing activity. Relative gas flux values measured within unit c are lower than observed for units a and b, for instance. This phenomenon might be a consequence of the permeability reduction or sealing processes due to the more advanced hydrothermal alteration (cf. Heap et al. 2019). This proposition is supported by the strongest changes in mineralogical and bulk geochemical composition observed in our samples, implying mineral (re-) precipitation is a major process in this particular unit.

Diffuse aureoles (unit a/b) on the eastern side of the fumarole field cover several thousand m² each, more than the area covered by HTF. The diffuse aureoles contribute 6 to 7% each to the total thermal energy release, equivalent to 25% of the energy emitted in the high-temperature zone (HTF + unit c). Their bleached surface, increased surface temperatures (Figure 4), and higher gas flux values (Figure 6 E-G) highlight their importance for the surface gas-drainage capability. However, mineralogical and bulk-chemical data suggests that the degree of alteration is less in these units, compared to unit c. This observation suggests that these sections are younger and therefore less altered and highlights their importance for the present-day degassing system.

Units d and e are large diffuse degassing domains of the western fumarole field of which unit d is a part of the thermal aureole surrounding F0 and unit e is a ~200 m long branched anomaly, located rim parallel west of F0. Both have a similar contribution to the total thermal energy release as units a and b. Unit d is separated from F0 by a Low-Temperature Zone (LTZ 3). The

transition from LTZ 3 to unit d is sudden and accompanied by a temperature jump of ~20 °C. The difference in the average temperature between unit d and LTZ 3 is on the order of 5 °C. Also here we observe apparent surface sealing for the whole central fumarole field. Unit f, the northern section of the F0 aureole, and anomalies in the area north of the fumarole field (unit g) have a minor contribution.

The Low-Temperature Zones (LTZ) 1 - 3, which separate diffuse aureoles from the HTF, have a Type 3 surface coloration, significantly lower temperatures and radiant flux and exitance values than the neighboring aureole regions, what indicates processes of surface sealing. Indeed, no gas flux could be constrained for the LTZ of the eastern fumarole field. From field observations and lab analyses (Figure 7), we constrained the LTZ as strong, sulfur-rich surface complexes that effectively seal the surface and inhibit gas escape. The depth of these sealed complexes can not be constrained by our data, but we can approximate the spatial extent. Considering only the 3 LTZs of the central and eastern fumarole field, they cover a total area of ~12,000 m², which is a significant fraction of the ALTZ. In other words, ~20% of the surface of the ALTZ is apparently sealed, which forces lateral gas flow to the aureole regions. This was proven by observations during the 2021 volcanic crisis at La Fossa. While at fumarole sites and diffuse active units like units a and b showed an increase in mean temperatures and thermal energy release, the radiation within LTZ remained stable and low, highlighting the efficiency of the proposed seal. This was observed for all central LTZs, and is exemplified in a cross-section through the eastern fumarole field section in Figure 11D.

## 6 Conclusion

Our investigation of the fumarole field of the La Fossa cone allowed us to constrain the degassing and alteration structure, define major, so far undescribed units of activity, and quantify their importance for the degassing system. Such high-resolution studies can greatly contribute to the understanding of structural architecture and add to our understanding of the intrinsic complexities of fumarole fields. This realisation has implications for hydrothermal alteration studies, particularly for the identification of local variability, since local variations are frequently associated with mechanical, chemical, and permeability contrasts. The recognition of such contrasts is of use for an improved assessment of volcanic and degassing activity, but also possibly for other hazard aspects, such as e.g. stability assessments. We anticipate that combined remote sensing and petrological studies will prove beneficial for pre-site reconnaissance surveys for hydrothermal energy exploration, the detection of sampling locations for alteration-related studies, and, importantly, for hazard monitoring of volcanic crater areas and associated risk assessment.


**7 Tables**
**Table 1:** Overview of the processed data sets that were used for the following analyses. From the optical data, an orthomosaic
and DEM were generated covering 3.74 km² with pixel resolutions of 8.6 x 8.6 to 17.3 x 17.3 cm. From the high-altitude
infrared overflight, an infrared mosaic was acquired covering 3.23 km² with 38 x 38 cm resolution. All data sets cover the
complete central section of the La Fossa cone.

| Data set | Acquisition date | Pixel resolution in cm | Coverage in km² | Point density in p/m² |
|---|---|---|---|---|
| 2019 orthomosaic | 14.11.2019 | 8.6 x 8.6 | 3.74 | 135.20 |
| 2019 DEM | 14.11.2019 | 17.3 x 17.3 | 3.74 | 33.41 |
| 2018 IR mosaic | 15.11.2018 | 38 x 38 | 3.23 | 5.64 |
















**8 Figures**

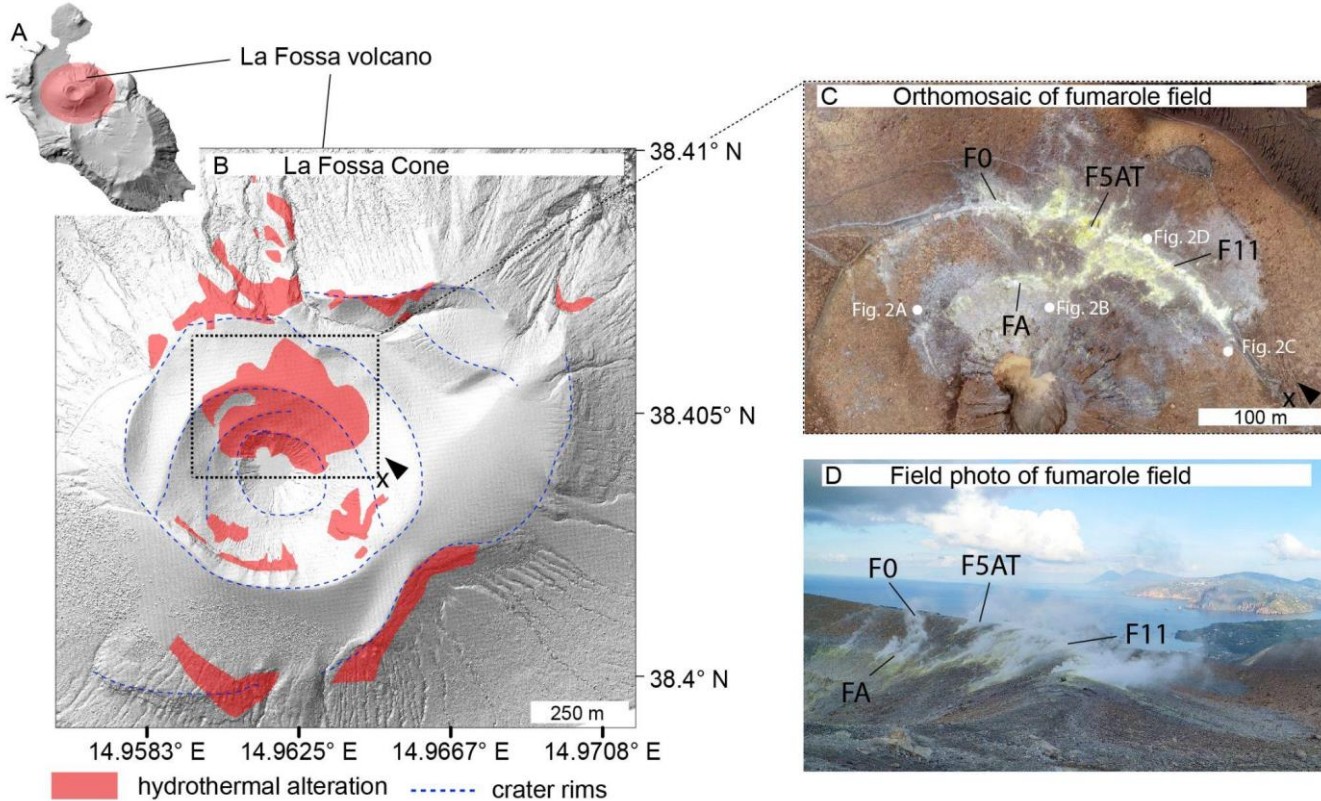


Figure 1: Overview of the degassing sites at La Fossa cone, Vulcano Island (Italy). A) Vulcano Island as a shaded relief map.
The red circle indicates the location of the La Fossa cone. B) Central summit area of the La Fossa Cone. Blue dashed lines
indicate crater rims from different eruptive episodes. Areas of degassing and hydrothermal alteration are highlighted in red
following Müller et al. (2021). The dashed box outlines the most prominent center of degassing and alteration, the high-
temperature fumarole field. C) Birds-eye view of the high-temperature fumarole field with prominent fumaroles F0, F5AT,
F11 and FA marked. The locations of field photographs of Figure 2 are indicated by white dots. D) Field photo of the fumarole
field. Location and viewing direction are indicated by an x and an arrow (B/C).

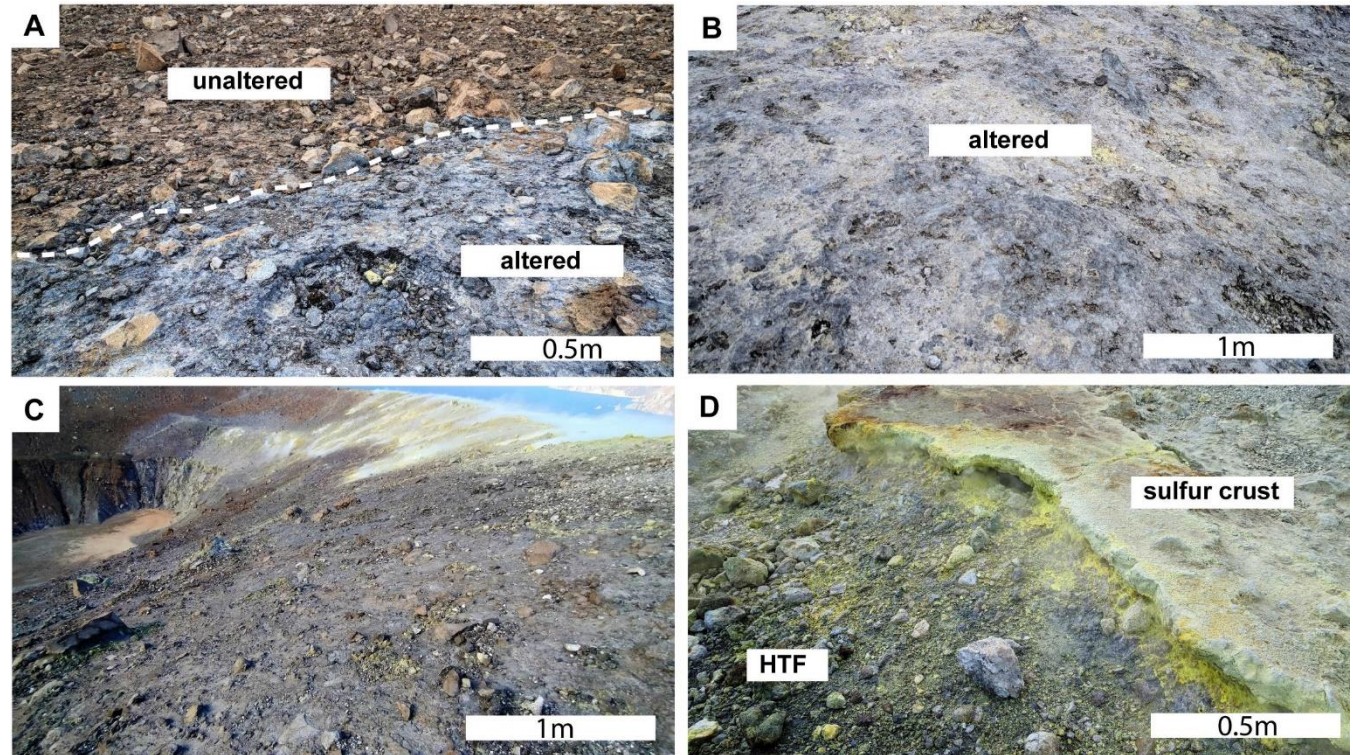


Figure 2. Different surface types and colorations at the La Fossa cone. A) Transition from unaltered to altered bleached
surface. B) Intensely altered and bleached surface. C) View from the east onto sealed surfaces. D) High-temperature fumarole
(HTF) and deposited sulfur crust.














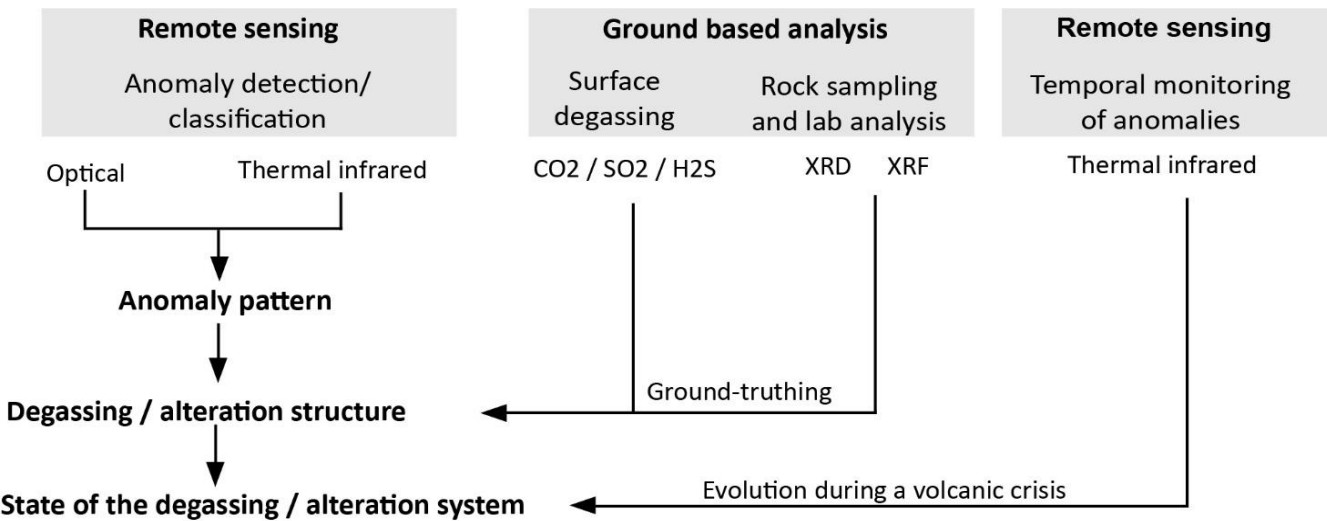


Figure 3: Overview of the general workflow used for this study. An anomaly detection from optical and thermal infrared
remote sensing data allows us to reveal the anomaly pattern and infer the surface structure of the degassing and alteration
system. To validate the observed structure, the remote sensing study was complemented by surface degassing measurements
revealing the present-day degassing pattern, and by X-ray diffraction (XRD) and X-ray fluorescence (XRF) analysis of selected
rock samples to prove different alteration units based on changes in mineralogical and bulk-chemical composition. Continuous
monitoring by high-resolution thermal remote sensing data allows to record dynamics within the system and to draw
conclusions about the general condition of the degassing/alteration units, e.g. with regard to alteration-related processes like
surface sealing.


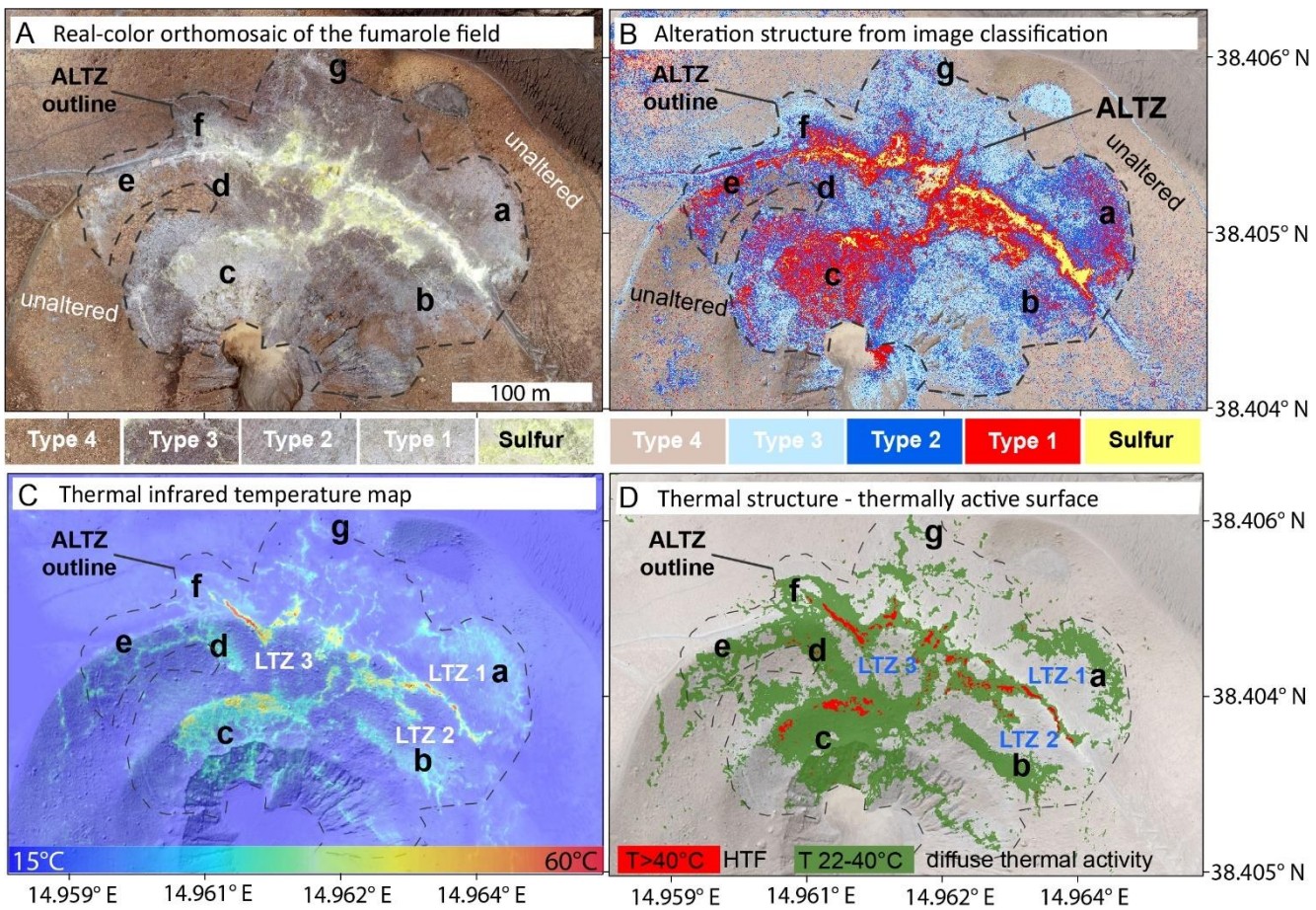

Figure 4: Alteration structure of the La Fossa fumarole field.  A) True color image of the high-temperature fumarole field with color samples of the surface types 1 - 4 and sulfur at the bottom of Fig. A.  B) Alteration structure of the fumarole field as revealed by PCA and image classification, represented by the classified surface types 1 - 4 and sulfur at the bottom of Fig. B. C) Thermal infrared temperature map of the fumarole field.  D) Simplified thermal structure of the fumarole field highlighting high-temperature fumarole location in red (T > 40 °C) and diffuse thermal activity in green (T = 22 - 40 °C). The dashed line labeled ALTZ outline demarks the boundary of visible optical effects at the surface and is referred to by us as ALTZ (Alteration Zone). The labels a-g demark notable large-scale anomaly units that can be observed in both, the optical data and thermal data. LTZ 1 - 3 demark Low-Temperature Zones that separate the high-temperature fumaroles and diffuse active units and cover significant parts of the central fumarole field. Note that the contrast of the background image has been reduced for highlighting in Subfigures B and D.

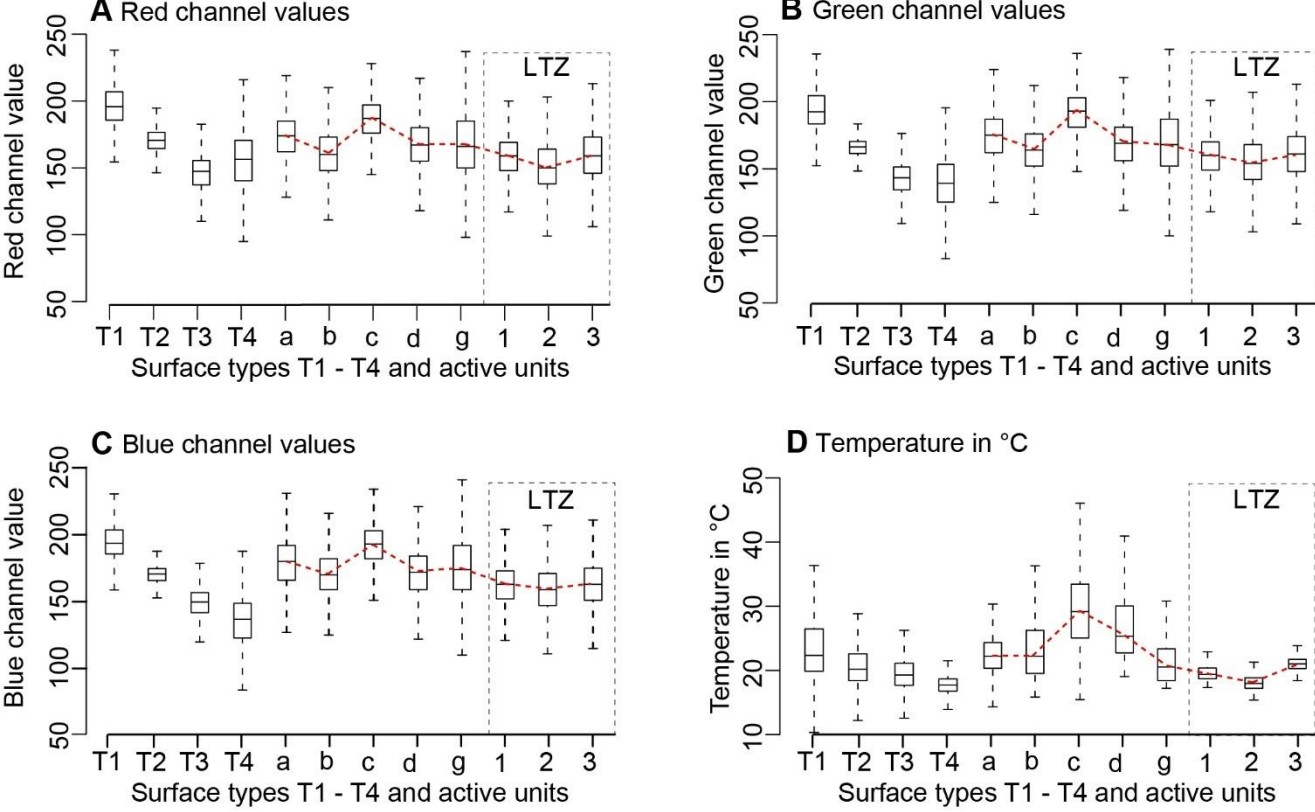

769

Figure 5: Boxplots of RGB color value- and temperature distributions observed for the different surface types 1 - 4 (T1 - T4),
identified active units a - g, and associated Low-Temperature Zones LTZ 1 - 3. Surface types and locations of identified units
are depicted in Figure 4B/D. A) Red channel value distribution. B) Green channel value distribution. C) Blue channel value
distribution. D) Temperature value distribution. Values are based on an analysis of 6.8 million pixels within the ALTZ. Both,
the optical (Figure A - C) and thermal (Figure D) value distributions show similarities with generally decreasing values from
the T1 - T4 surface, a peak in unit c, and low values for LTZ 1 - 3.

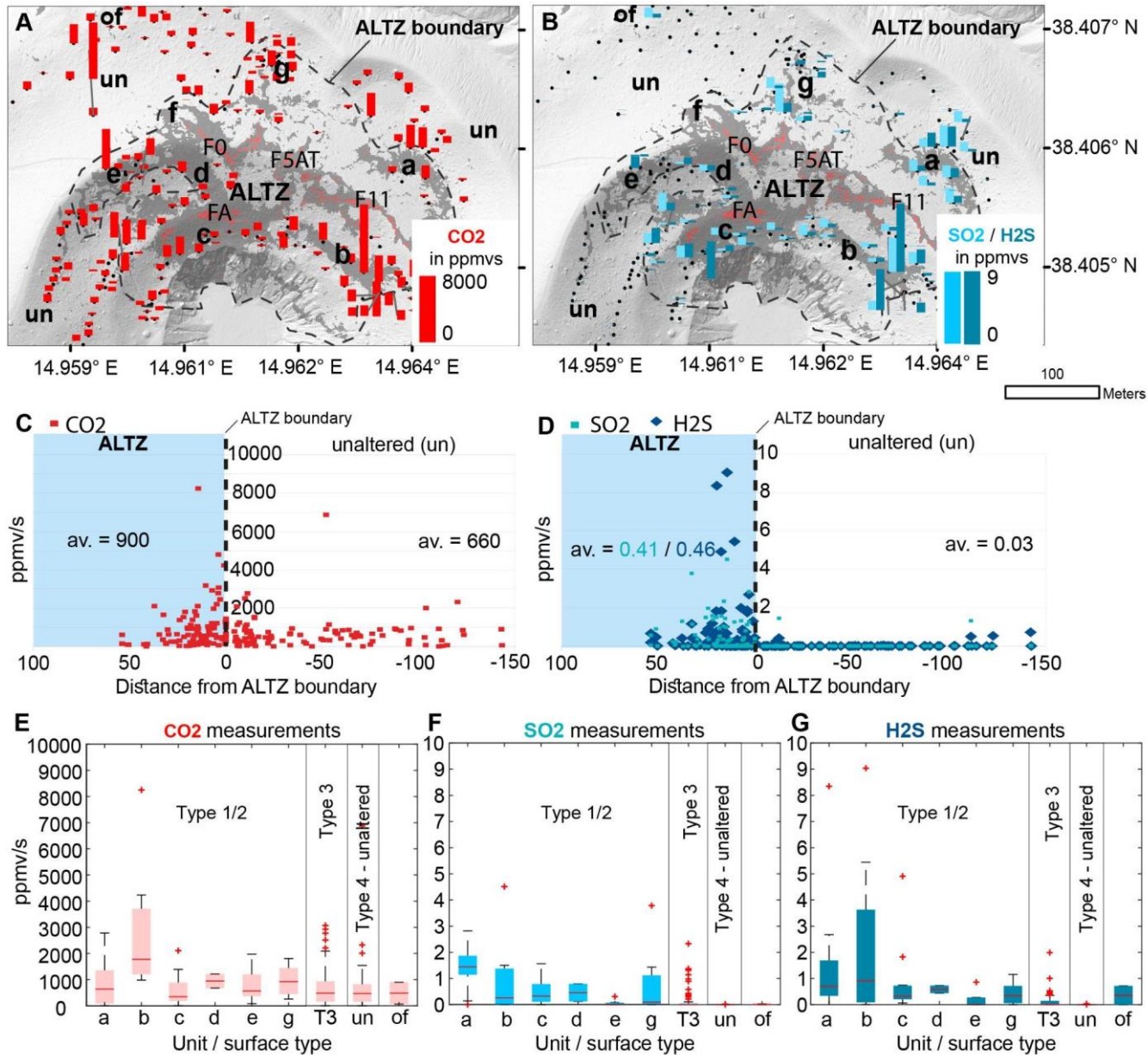

Figure 6: A/B) Spatial distribution and flux values for $CO_2$ (red bars in A), $SO_2$ (light blue bars in B), and $H_2S$ (turquoise bars in B) in a map view for 200 measurement points. Each bar represents a relative flux value at a measurement location. In case no flux was detectable, the respective location is marked with a black dot only. The dashed line highlights the ALTZ (Alteration Zone) boundary. Dark grey features in the background highlight the thermally active surface (compare Figure 2D). Labels F0, F5AT, F11, and FA mark prominent high-temperature fumaroles. The labels a - g mark notable large-scale anomaly units. C/D) Flux values are plotted by distance to the ALTZ boundary (dashed line). Measurement points within the ALTZ are represented by positive distances from the ALTZ boundary (highlighted by blue background) and measurements outside the

ALTZ by negative distances (unaltered). A generally higher flux is observed within the ALTZ, but while $CO_2$ is also abundant
outside the ALTZ, significant $SO_2$ and $H_2S$ fluxes were observed exclusively within the ALTZ, especially on the outer edges
and associated with units a - g. The averaged flux values (av) are depicted in the respective sections of C and D.  E-G) Relative
flux values of identified units (a - g), Type 3 surface (T3), the unaltered surface (un), and fumaroles on the northern rim at a
distance (of) highlight the spatial variation of different gas species with high flux values for units a and b, lower flux values
for e.g. unit c and generally lower flux of sulfuric gas species in the unaltered regime outside the ALTZ.

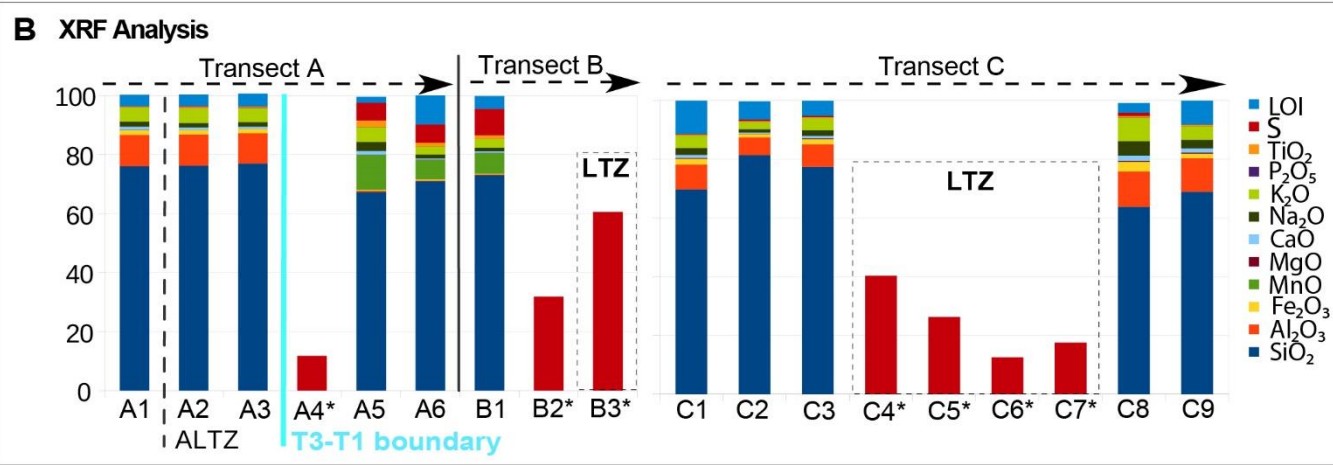

Figure 7: Mineral and bulk chemical composition of rock samples along 3 transects A - C, crosscutting alteration gradients, and structural units. A) Overview map with defined surface types T 1 - 4, highlighting the area with visible optical changes at the surface, referred to as ALTZ (Alteration Zone, marked by a dashed line). The three transects A - C were placed so that they crosscut prominent units. Mineralogical compositions from XRD (X-ray diffraction) are depicted by circular plots at the bottom of Fig. A.  B) Bulk chemical composition from XRF (X-ray fluorescence) analysis of transects A - C. Transect A/B) With increasing alteration intensity we observe a relative decrease of the initial mineral phases sanidine and cristobalite whereas the sulfur content increases. Note that the mineral composition in this figure is normalized to 100% non-amorphous minerals. In the chemical composition, we observe a significant decrease of $Al_2O_3$ and $Fe_2O_3$ but an increase of MnO, $TiO_2$,

and S with increasing alteration, especially at the Type3 - Type1 (T3 - T1) boundary marked in light blue. For transect C we observe a dominant increase of S (17 - 40%) for samples taken within the LTZ (Low-Temperature Zone). Compared to other transects, changes in $Al_2O_3$, $Fe_2O_3$, $TiO_2$, and MnO are less significant. For samples marked with an asterisk (*) XRF results are not available. No XRD results are available for transect C.

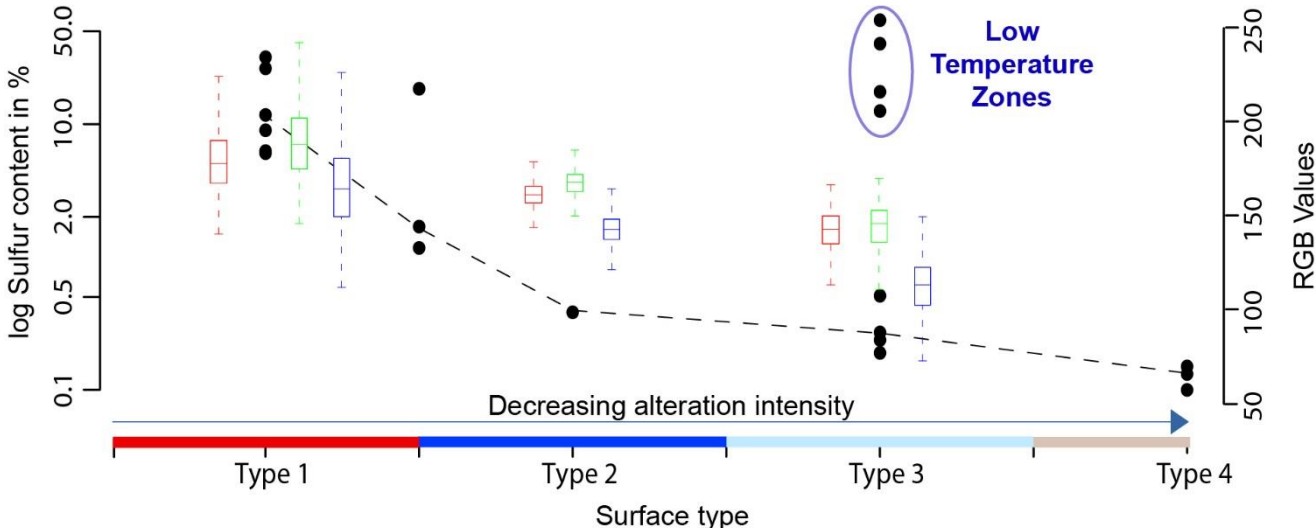

Figure 8: Relation of inferred surface type (T 1 - 4) and sulfur content of rock samples taken in the respective surface T 1 - 4. Black dots mark the sulfur contents of rock samples and are shown on a log scale versus the surface type from which the sample was taken, labelled by Type 1 - 4 on the x-axis and demarked with boxes using the same color code as throughout the manuscript. With decreasing alteration from Type 1 surface to Type 4 surface, we see a significant decrease of sulfur content from up to 100% for Type 1 to < 1% for Type 4 surface. The black dashed line illustrates this. An exemption is the Type 3 surface where we observe two distinct clusters, one with low sulfur values and one with exceptionally high sulfur values. These high sulfur values belong to samples taken in the LTZ (Low-Temperature Zones), which separate fumaroles from the larger diffuse active units a and b (compare Figure 4) for instance. These samples indicate that LTZ represent sulfur-rich crusts that block heat and gas flux from the surface. The colored boxplots show the distribution of RGB values for the respective surface Type 1 - 4, showing a similar trend of decreasing values. Red boxplots represent the red value, blue and green the respective blue and green channel values of the image. The coincidence between both indicates a direct relation or control of sulfur onto the surface colorization.

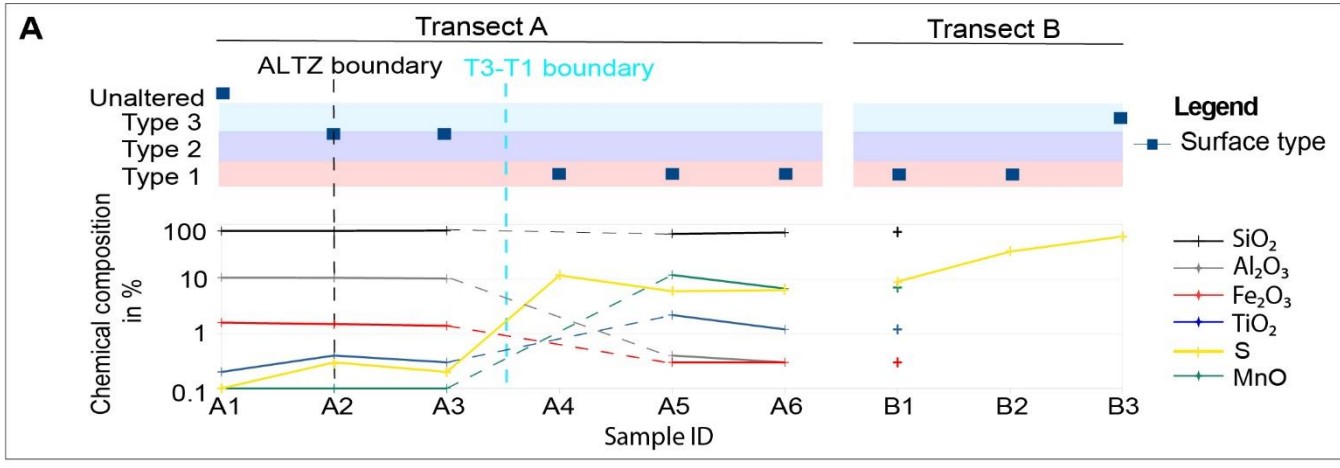

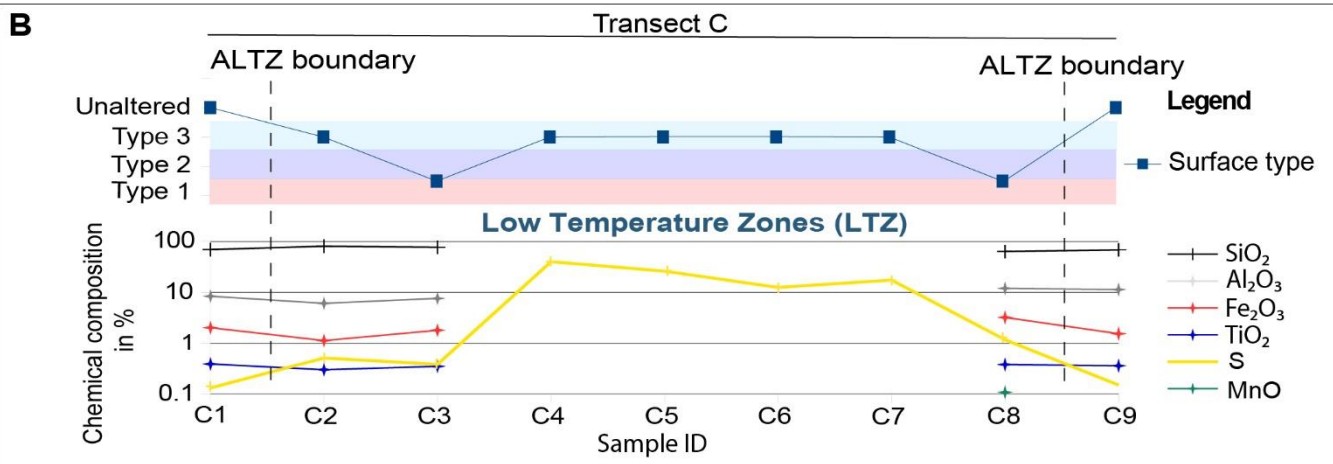

Figure 9: Changes of surface type and bulk-chemical composition observed along transects A - C. Locations for the transects and sample ID are shown in Figure 7. A) With increasing alteration from Type 4 to Type 1 surface we observe a reduction of $Fe_2O_3$ and $Al_2O_3$ and an increase in sulfur. While changes observed at the ALTZ boundary (black dashed line) are only minor, strong changes are observed at the Type 3 - Type 1 boundary (T3 - T1, blue dashed line). Sulfur contents in transect B were so high that XRF results were only available for sample B1.  B) Changes observed in the eastern fumarole field along transect C are less significant, with the exception of extraordinarily high sulfur content for Type 3 samples collected in the LTZ.

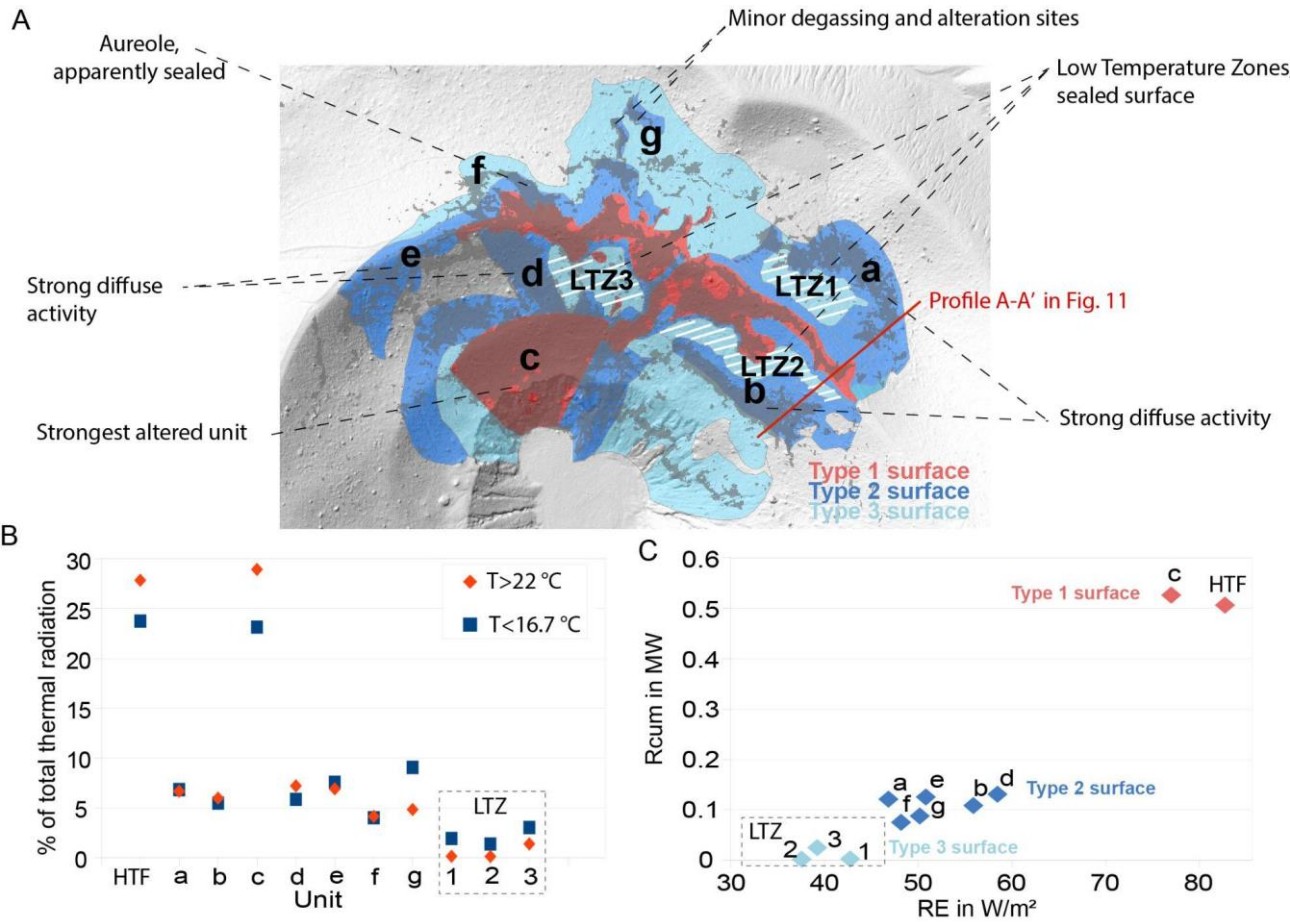

827

Figure 10: Anatomy of the fumarole field. A) Simplified structure of the fumarole field highlighting surface types and structural units of increased diffuse activity (a-g) or areas of apparent surface sealing (LTZ 1 - 3 marked by white lines). B) Contribution to the total thermal radiation in % for HTF (high-temperature fumaroles), units a - g, and LTZ 1 - 3 (Low-Temperature Zones) considering pixels with T < 22 °C (blue) and for identified units based on a spatial constraint and pixel temperatures > 22 °C (orange). C) Radiant exitance (RE) in W/m² and cumulative radiation (Rcum) in MW. Rcum is the cumulated background corrected radiant exitance (Equation 2 in the method section) of all pixels associated with the respective active unit. Note that for Rcum only pixels with T > 22 °C were used. We can clearly distinguish different thermal regimes that are also coincident with surface types identified in the optical data.



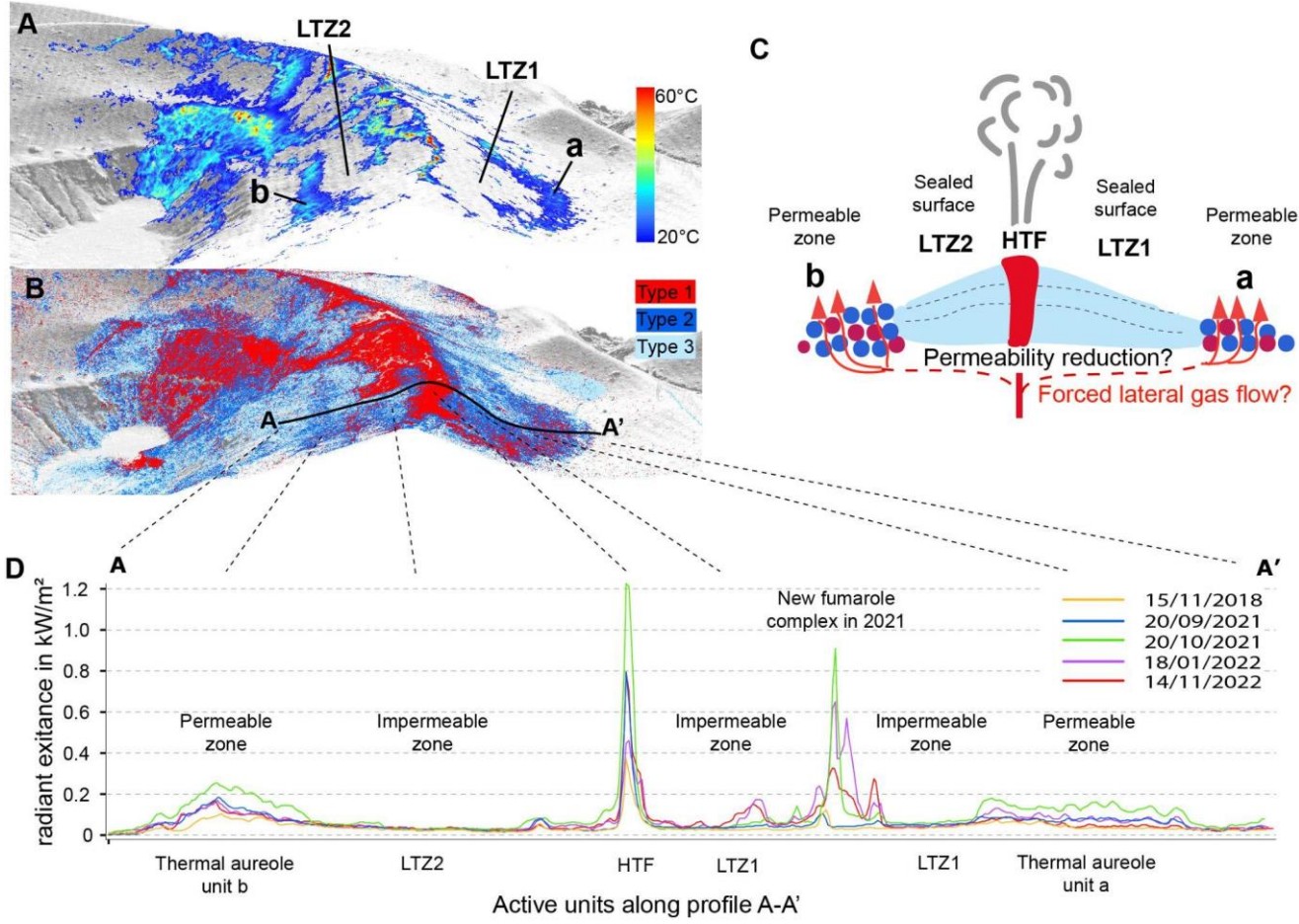


Figure 11: Cross section of the eastern fumarole field along Profile A - A' (location see Fig.10) highlighting the structural
setup from high-temperature fumaroles in the center to LTZ and diffuse aureoles at a distance. A) Thermal structure along the
cross-section. B) Alteration structure along the cross-section. C) Schematic sketch along cross-section A-A', highlighting the
central LTZ that might be controlled by surface sealing processes or deeper effects of permeability reduction in the vicinity to
the high-temperature fumaroles due to long-term gas-rock interaction and alteration processes. D) Evolution of thermal
radiation values during a volcanic crisis. While thermal radiation at fumaroles and aureoles increased, radiation values of LTZ
remained unchanged, therefore highlighting the efficiency of surface sealing.






**9 Appendices:**
**Appendix A: RGB value distribution of defined surface types 1 - 3 within the ALTZ**

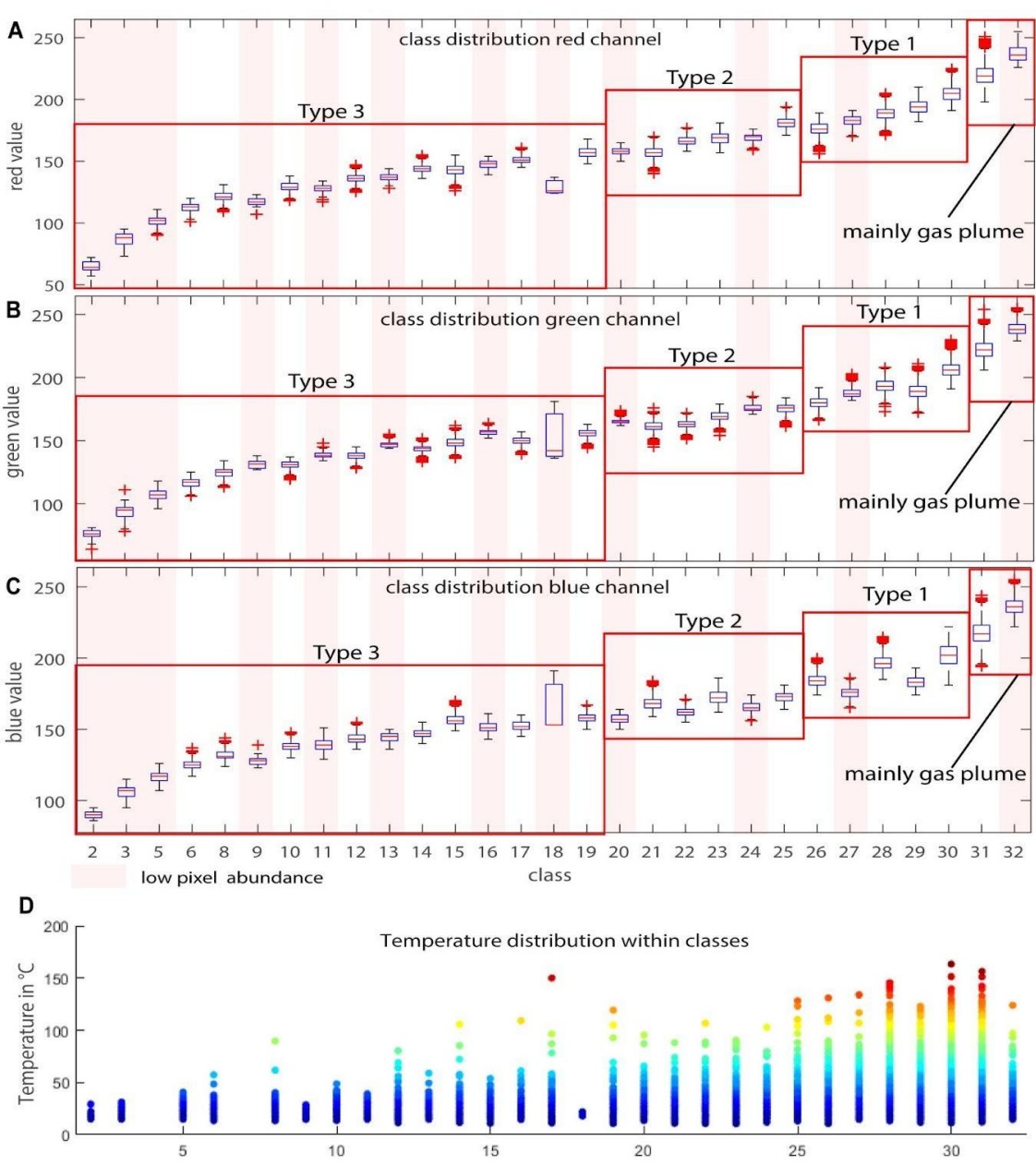

Figure A1: Boxplots of RGB value distribution for the defined surface Types 1 - 3. Classes (unsupervised classification 32
classes) marked with the transparent red bar only have minor pixel abundances. Red boxes depict the spectral range of Type 1
- 3 surfaces. Class 31 and 32 are mainly associated with the fumarole steam plume.


**Appendix B: Gas measurement procedure - simplified accumulation chamber approach**

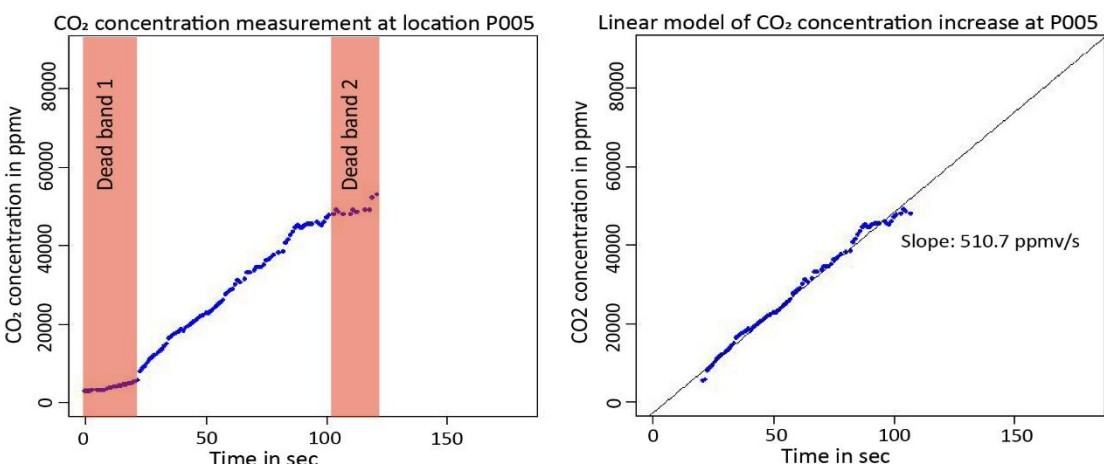

*Figure B1: CO₂ measurement at location P005, here shown representative for all measurement points. Dead bands at the beginning and end of the measurement were removed and the intensity of gas flux was characterized by linear regression through the constantly ascending part of the graph. The slopes of the linear model allow a relative comparison of single measurement points.*

To compare the observations from remote sensing to present-day surface degassing, gas-measurement campaigns were performed in September 2021 and November 2022. The surface degassing was measured at 200 points within the northern part of the La Fossa cone (Figure 6 in the main manuscript), in a simplified Multigas accumulation chamber approach.

The simplified accumulation chamber consists of the measurement unit, a Dräger X-am 8000, coupled to a 10.3 cm diameter and 16.5 cm long plastic chamber by a 116 cm long tube with an inner diameter of 0.5 cm, resembling a simplified accumulation chamber. The plastic chamber has a volume of 1374.8 cm³, and the tube has a volume of 91.1 cm³ so that the total system volume is 1465.934 cm³. The pumping rate is 0.35l per minute. The plastic chamber was equipped with an open valve which was a necessity as the Dräger is an actively pumping system. Therefore, concentration increases in the chamber can be considered as surface flow and as independent from pumping effects. The Measurement unit is protected by a preceding 2µm filter, preventing dust and vapor from entering the unit. Note that we use Flux values in this study only for relative comparison and detection of the spatial variability of certain gas species and flows. A precise flux estimate is beyond the scope of this publication and can not be constrained as we did not measure gas temperatures and humidity at sampling locations.

The measurement unit, a Dräger X-am 8000 handheld Multigas device was equipped with 6 sensors measuring $CO_2$, $CH_4$, $SO_2$, $H_2S$, $H_2$, and $O_2$ simultaneously. The relevant species for this work are $CO_2$, $SO_2$, and $H_2S$, therefore only these will be considered in detail. The $CO_2$ sensor is a Non-Dispersive Infrared (NDIR) sensor. NDIR sensors use the absorption characteristics of $CO_2$ at ~ 4 µm, which leads to a concentration-dependent amplitude loss of the internally emitted IR light. The sensor has a detection threshold of 0.01 vol% $CO_2$ and is calibrated for measuring $CO_2$ in a range of 0-5 vol% at a

resolution of 50 ppm under normal ( -20 - 50 °C, 10 - 95% RH, and 700 - 1300 hPa) atmospheric conditions. The response
rate is < 10 sec for reaching T50- and < 15 sec for reaching T90 concentrations. The $H_2S$ sensor is an electrochemical sensor
with a detection limit of 0.4 ppm and a resolution of 0.1 ppm, measuring in a range of 0 - 100 ppm $H_2S$ under normal
atmospheric conditions. The response time for T90 values is > 15 seconds and the accuracy of the measurement is +-5% of the
measured value. The $SO_2$ sensor is an electrochemical sensor with a detection limit of 0.1 ppm and a resolution of 0.1 ppm,
measuring as well in a range of 0-100 ppm under atmospheric conditions. The response time for T90 values is < 15 seconds
and the accuracy of the measurement is 2% of the measured value. The NDIR $CO_2$ sensor is robust against cross-sensitivities.
However, electrochemical sensors can be vulnerable to cross sensitivities ($SO_2$, $H_2S$, $Cl_2$), resulting in uncertainties of the
measurement of a few percent of the measurement value.
The approach of combining the Dräger Multigas with an accumulation chamber was developed and adapted as a consequence
of uncertainties encountered in previous campaigns. The different sensors have slightly different reaction times for ascending
gas concentrations and significantly different reaction times for descending gas concentrations. Comparing sensor readings
directly, therefore, might lead to odd gas ratios. For that reason, instead of the direct gas readings, we use the slope of the
ascending gas concentration within the accumulation chamber to produce more reliable estimates of the surface flow.
For a relative comparison of degassing rates of the single measurement points, the gas data was plotted and the representative
part of the graph, resembling a constantly ascending slope, was used to calculate the concentration increase by linear regression.
Data points of the "Dead Bands" at the beginning and end of each graph were removed. In this way, we achieve a relative gas
flux from surface that allows us to analyze spatial variations of gas flux throughout the study area. An overview of all gas
measurement points will be given in Figure 6. The aim of the gas measurements was not to provide accurate flux estimates but
to highlight and quantify the spatial variability of the surface flux of certain gas species.
Each measurement was performed under similar conditions. Locations were selected in a way that they represent similar
surface conditions, considering a spatial distance to fumarolic vents and an unsealed surface, for instance. Measurement
locations typically were small areas with a naturally "open surface", often embedded in broader areas of the sealed surface.
Such spots typically can be identified by loose gravel on the surface and in case slightly different coloration. For the
measurement, the surface was cleaned and gravel was removed to provide a flat contact surface. Then the measurement was
started, and the plastic chamber was placed on the ground and sealed on the bottom with fine-grained material. The average
measurement duration was 2 min. In case of very rapidly ascending $SO_2$ or $H_2S$ gas concentrations, the chamber was removed
from the ground before and the system was flooded with fresh air to protect the sensors from critically high acid concentrations.
This procedure was chosen to ensure a fresh air flooded chamber at the beginning of each measurement and to record the initial
background gas concentration. Further, it allows better identification of the measurement start- and end-points within the
respective data sets, as each data series has two dead bands, one at the beginning and one at the end. Figure B1 shows a typical
graph of a $CO_2$ measurement, with the Dead-Band at the beginning and end of each measurement and the constant ascending
graph, representing the gas concentration increase within the chamber. The "Dead Bands" represent parts of the measurement
where the accumulation chamber was placed on the ground but not sealed yet, or removed from the soil at the end of the
measurements. Dead Bands at the beginning of the measurement were typically on the order of 20 - 30 s.

 **Appendix C: Detail views on identified units**

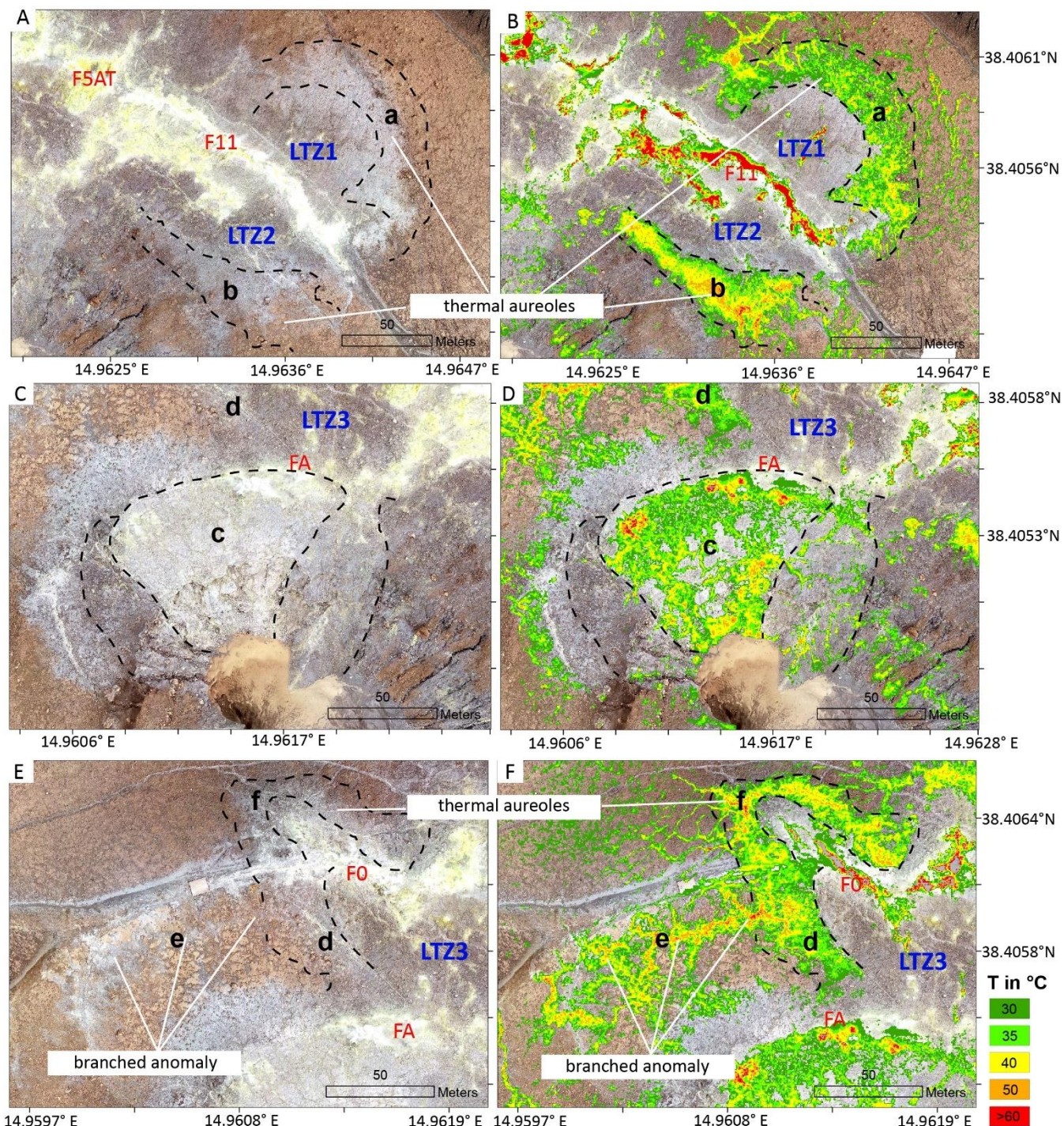

Figure C1: Detail views of distinct units a - f and LTZ 1 - 3 in a true color representation as seen from our 2019 orthomosaic
data and an overlay by the thermal data with T > 30 °C. A/B) Shown are units a and b and respective LTZ 1 and 2. Note the
outward spatial offset of both thermal units with respect to the surface coloration. C/D) Unit c is characterized by a network
of thermal anomalies embedded in the colder surroundings. E/F) Thermal aureole d and f, and branched anomaly e. Also here
an outward shift of the thermal feature with respect to the surface coloration is observed, which could indicate gradual sealing
processes with proximity to the main vents.
**Appendix D: Spearman correlation test for non-normal distributed variables**
The test for correlation between optical and thermal anomalies was performed using the ggpubr package (Kassambara, 2019)
in the statistical software environment R. The method used was Spearman's rank correlation which is suggested to be used for
non-normal distributed data. The correlation test is based on the vectorized classification raster data set (classp_fumclip) with
8,890,830 data points with the analyzed variables pixel class (0 - 32) and pixel temperature (20 - 150 °C). The results show a
correlation factor of 0.3485299, which is considered a mean positive correlation, and a p-value of 2.2e-16 proves statistical
significance.
Spearman's rank correlation rho
data: x and y
S = 7.6277e+19, p-value < 2.2e-16
alternative hypothesis: true rho is not equal to 0
sample estimates:
rho
945 0.3485299

**Appendix E: Thermal aureoles and LTZ indicated in field photographs**

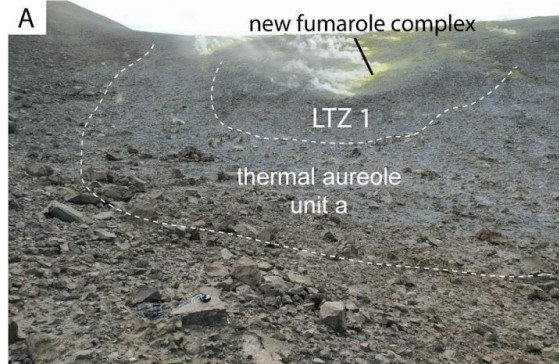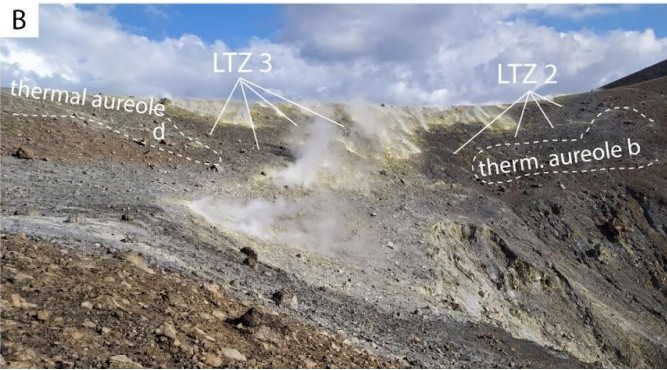


Figure E1: Thermal aureoles and Low-Temperature Zones (LTZ) depicted on field photographs. A) Thermal aureole a and
LTZ 1.  B) Thermal aureole b and d with LTZ 2 and LTZ 3.
**Appendix F: XRD results of samples taken along transects A and B**
Table F1: XRD results of samples taken along transects A and B

| Sample ID | A1 | A2 | A3 | A4 | A5 | A6 | B1 | B2 | B3 |
|---|---|---|---|---|---|---|---|---|---|
| **Sanidine** | 86.7 | 85.7 | 87.4 | 61.5 | 72.2 | 60.3 | 68.2 | 49.2 | 0 |
| **Cristobalite** | 13.3 | 14.3 | 12.6 | 18.2 | 0 | 0 | 0 | 0 | 0 |
| **Coesite** | 0 | 0 | 0 | 0 | 0.7 | 0 | 0 | 0 | 0 |
| **Sulfur** | 0 | 0 | 0 | 20.3 | 25.1 | 39.7 | 31.8 | 50.8 | 100 |
| **Amorphous** | 50 | 0 | 0 | 0 | 0 | 50 | 50 | 0 | 0 |

**Appendix G: XRF results of samples taken along transects A-C.**
Table G1: XRF results of samples taken along transects A-C. Note that samples with S > 10 % were not analyzed by XRF.

| S-ID | $SiO_2$ (%) | $TiO_2$ (%) | $Al_2O_3$ (%) | $Fe_2O_3$ (%) | MnO (%) | MgO (%) | CaO (%) | $Na_2O$ (%) | $K_2O$ (%) | $P_2O_5$ (%) | LOI (%) | S Eltra (%) |
|---|---|---|---|---|---|---|---|---|---|---|---|---|
| A1 | 76.0 | 0.2 | 10.7 | 1.6 | 0.1 | 0.1 | 1.1 | 1.8 | 4.8 | 0.1 | 3.8 | 0.1 |
| A2 | 76.2 | 0.4 | 10.6 | 1.5 | 0.1 | 0 | 0.8 | 1.5 | 5.0 | 0.1 | 3.9 | 0.3 |
| A3 | 77.0 | 0.3 | 10.2 | 1.4 | 0.1 | 0 | 0.7 | 1.6 | 4.8 | 0.1 | 4.3 | 0.2 |
| A4 | x | x | x | x | x | x | x | x | x | x | x | 11.7 |
| A5 | 67.3 | 2.2 | 0.4 | 0.3 | 11.9 | 0.1 | 1.2 | 3.1 | 5.0 | 0.1 | 2.0 | 6.0 |
| A6 | 71.0 | 1.2 | 0.3 | 0.3 | 6.7 | 0.0 | 0.4 | 1.3 | 2.7 | 0.0 | 9.8 | 6.3 |
| B1 | 73.0 | 1.2 | 0.3 | 0.3 | 7.0 | 0.0 | 0.5 | 1.3 | 3.0 | 0.0 | 4.3 | 9.0 |

| | | | | | | | | | | | | |
|---|---|---|---|---|---|---|---|---|---|---|---|---|
| B2 | x | x | x | x | x | x | x | x | x | x | x | 31.9 |
| B3 | x | x | x | x | x | x | x | x | x | x | x | 60.5 |
| C1 | 69.7 | 0.4 | 8.4 | 2.0 | 0 | 0.3 | 1.0 | 2.4 | 4.2 | 0.1 | 11.3 | 0.1 |
| C2 | 81.3 | 0.3 | 6.1 | 1.1 | 0 | 0.1 | 0.4 | 1.1 | 2.5 | 0 | 6.1 | 0.5 |
| C3 | 77.4 | 0.4 | 7.6 | 1.8 | 0 | 0.2 | 0.9 | 2.0 | 4.0 | 0.1 | 5.1 | 0.4 |
| C4 | x | x | x | x | x | x | x | x | x | x | x | 40.3 |
| C5 | x | x | x | x | x | x | x | x | x | x | x | 26.2 |
| C6 | x | x | x | x | x | x | x | x | x | x | x | 12.5 |
| C7 | x | x | x | x | x | x | x | x | x | x | x | 17.5 |
| C8 | 63.8 | 0.4 | 12.0 | 3.2 | 0.1 | 0.4 | 1.7 | 5.0 | 8.0 | 0.1 | 3.2 | 1.2 |
| C9 | 68.9 | 0.4 | 11.4 | 1.5 | 0 | 0.4 | 1.3 | 3.0 | 4.7 | 0.1 | 8.1 | 0.2 |

## 10 Data availability

The remote sensing data used for this study and relevant processing steps are published in a Zenodo data repository https://doi.org/10.5281/zenodo.12586672. Other data will be made available on request.

## 11 Author contributions

D.M conceptualized the study, collected data, performed the remote sensing and gas analysis, and led the manuscript writing. T.R.W. provided funding, supported the conceptualization, and supervised the writing. V.T. supported the conceptualization, performed XRD analysis, and supervised the writing. J.S. performed XRF analysis and supported the writing. A.K. performed XRD analysis and supported the writing. E.D.P. collected data and samples, supported all field works and the writing of this manuscript. A.F.P. supported fieldwork and on the ground logistics, acquired data, and supported the writing. M.Z. supported the gas measurement campaign and supported the writing. B.D.J. supported the fieldwork and writing of this manuscript.

## 12 Competing interests

The authors declare that they have no conflict of interest.

## 13 Acknowledgements

We are grateful for the financial and material support provided to realize this study. This work is contributing to the focus site Etna and was financially supported by GFZ Potsdam. Financial support to realize this study was also provided by DAAD research grant Nr. 57556282 and by ERC project 'ROTTnROCK*, a research project funded by the European Research Council under the European Union's Horizon Europe Programme / ERC synergy grant n. [ERC-2023-SyG 101118491]. We furthermore thank INGV Palermo for collaboration and support, especially during the 2021 crisis, without which parts of this study could not have been realized.

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
