# Peer review of "Anatomy of a fumarole field; drone remote sensing and petrological"

_EGUsphere, 2023_

## Referee Comment (RC1)

Dear Editor,

Please find below my report on the manuscript "Anatomy of a fumarole field; drone remote sensing and petrological approaches reveal the degassing and alteration structure at La Fossa cone, Vulcano Island, Italy" (egusphere-2023-1692) by Daniel Müller et al.

The present manuscript presents a combination of visible and thermal imagery, mineral and geochemical analyses of rock samples and CO2 soil degassing data with the aim of classifying and quantifying the alteration and degassing structures at the Vulcano Fossa volcano. The manuscript uses a number of novel techniques, with the main emphasis on the classification of thermal and visible images (orthophotos), similar to previously published work (Müller et al., 2021), with the addition of mineralogical and geochemical analyses of hydrothermally altered samples. Whilst this approach is promising, and the results given are interesting, I think that some of the methods employed need much more extensive descriptions. Some of the techniques employed may also be flawed or, at best, misunderstood. Lastly, I do not feel that the authorship is currently up to standard for publication, though the written English is fine in itself. Hence it is my opinion that the manuscript requires major revisions before it could be considered for publication. I provide some suggestions that the authors may chose to follow. Given that the revisions may be substatial, a review of the revised manuscript may be necessary. In this case I would be happy to act as referee should you require my services.

Best regards
David Jessop

**PCA and image classification**

Probably my major concern regarding the manuscript concerns the lack of description of the "Principal Component Analysis", (PCA) and the image classifications. My reasons are two fold:

1. The PCA and image classification processes, provided within the ArcGIS propitiatory software, are "black boxes" with no description of what is happening under the hood. As written, it is not easy to understand how these process work, or what data they provide. The interested (and reasonably competent) reader should be able to reproduce the results of this study and I don't think that is currently the case.

2. In order to reproduce these results, the interested reader would require access to the proprietary ArgGIS software which means paying 500-700€. I don't think that this is fair. Whilst this may not be an issue for wealthy universities, it could block researchers with less access to funding from utilising the approaches employed in the present study. A proper description of the methods could allow such researchers to look for cheaper of even open-source alternatives.

PCA consists of taking a multi-dimensional dataset and finding the orthonormal basis vector space that describes the data whilst minimising the variance along each vector basis (component). This is achieved by calculating by projection of the data onto a set of orthogonal axes where the variance of each data set is represented by the eigenvalues of the data. These eigenvalues are the principal components. This is usually achieved using a reduced singular value decomposition (SVD) which produces the eigenvalues of the dataset. PCA then takes the list of ordered eigenvalues which are typically used to perform dimensional reduction in high-dimensional data sets. This consists of rejecting any components that do not contribute significantly to the overall variance of the data. In

the present study this is applied to an RGB image (i.e. 3D) and the authors take 3 principal components, so there is no dimensional reduction and hence the "PCA" is kind of redundant.

Regardless of how we name this process, it is unclear what the ArcGIS algorithm produces in the "Principal Component" band images (cf. L220-223) – PCA gives only the variance. Please indicate how this information is used to transform the RGB image.

The image classification, named as "unsupervised classification", process is poorly documented. Indeed unsupervised classification is a blanket term for a multitude of different families of algorithms so, to be able to reproduce these results, one would have to know which algorithm was chosen and why. The choice of 32 classes seems to be completely arbitrary (50 were used in Müller et al., 2021) and, furthermore, the individual classes are then regrouped (see fig. A1 for example). Could a smaller set of classes (say 4) not have been used to obtain similar results? I strongly urge the authors to justify their choice.

Whilst the authors refer to previous work (Müller et al., 2021) as a source for their methods, but neither the PCA nor the classification strategies are sufficiently well described in that work either. Without having to detail the algorithms in their entirety, the authors should please sufficiently explain their methods in the present manuscript so that they can be followed with the aim of reproducing their results.

**Thermal image processing**

[Figure]

Fig. 1: "Brightness" temperature predicted by Planck's law compared to the scaling proposed by the authors (eq. 1). The two curves cross at a temperature of 383 K (about 90°C).

The authors state that they produce temperature maps from the 16-bit radiometric greyscale orthophoto using a linear mapping given by their eq. 1 using the radiometric resolution as a scaling factor. Owing to the non-linear behaviour of IR sensors this would seem unlikely to hold for more than a very limited range of greyscale values. Furthermore, it is typically necessary to use Planck's law which predicts the "brightness" temperature of an object from the intensity of incoming radiation registered by the sensor (i.e. radiometric value). By way of illustration, I have produced the above graphic (Fig. 1). Here we see that there is only one point of intersection for the two curves and the predicted temperatures can be drastically different. That said, I am not familiar with the FLIR Tau camera as used in this study and do not have access to the radiometric conversion factors necessary to correctly plot the curve for this camera. If it is like many other FLIR and other IR cameras that I have used, this information can be found in the EXIF (image metadata) which can be readily extracted using the ExifTool software, for example. However, the authors should check their data and any calculations that depend on the temperature. Some of the stated temperatures and thresholds (e.g. 40 °C for the "high-temperature fumaroles") are rather low given the vent temperatures recorded in other works (vent temperatures are well in excess of 100°C, cf. Diliberto, 2017; Mannini et al., 2019; Diliberto, 2021).

**Authorship and increasing the scope of the paper**

A quick survey of the the first dozen or so references showed that they were cited once in the introduction and nowhere else in the manuscript. The discussion contains only six references, and is often a rehash of the results section rather than a forum for putting these results into a fuller context and comparing them to previous works on Vulcano and, potentially, other volcanoes. Several references (e.g. Chiodini et al., 1996; Chiodini et al., 2005; Harris et al., 2009; Mannini et al., 2019) have made estimations of degassing and/or heat budgets with Harris et al. (2009), in particular, having made detailed descriptions of the fumarole field. I find it strange that the authors have not chosen to make the comparison with these works, particularly given the ongoing and unrest at Vulcano with the recent well-documented paroxysms. Fig. 11 identifies a structure with an increased radiant density and labels it as "new fumarole complex in 2021", but this is not discussed anywhere. This would be very important information for assessing the activity at this volcano. Curiously, Fig. 10 hints at a heat budget having being calculated, but this is not discussed in the manuscript. I note also that the area of the alteration zone (ALTZ), that is the area affected by hydrothermal activity, is given as 70 000 $m^2$ (note typo "770 000 $m^2$" on L301) which is very close to the "diffuse heated area" of 63 000 $m^2$ calculated by Mannini et al. (2019) using approximately contemporaneous data. Of course, I have my own professional biases in mentioning this, but the authors have already done the work so it is suprising that it is only mentioned in passing.

Concerning alteration of the edifice, there have been a number of studies in recent years trying to ascribe thermal properties of volcanic rocks to hydrothermally altered samples (typically andesite). These results may be interesting to discuss. See Heap et al., (2022) in particular. Section 5.3 briefly mentions the role of alteration and permeability:

"Relative gas flux values measured within unit c are lower than observed for units a and b, for instance. This might be a consequence of the dynamics of hydrothermal alteration and indicate permeability reduction or sealing processes due to the advanced state of alteration like proposed by Heap et al., 2019." (L592-594)

Furthermore, a sequence of alternating high and low permeability zones are identified in Fig. 11, but each result is only discussed independently and there is no real synthesis of the large and important set of results. This is one discussion point, in particular, that is really important for assessing volcanic hazard and I find it frustrating that this point has not been fully persued.

*A minima*, the figure captions should allow the reader to understand the figure in isolation and, currently, this is not the case for several of the figures. The main cuprits are

Fig 2 – give locations of each photo, also show these locations on Fig 1B.

3 – describe the grey blocks. Why are there two blocks for Remote Sensing?

8 – what are "RGB values", as only one value is given here. Is this for one band in particular or an average? Why are RGB values being used rather than one of the "PCA" image bands?

9 – What is the absicssa in this figure? How is it that Transect A blends into Transect B? I think it would be worthwhile to combine this figure with Fig. 7, particularly as one requires the locations of the transects to understand what is going on in this figure.

Other suggestions for the figure captions are:

4 – D, please give the thermal thresholds for each class.

5 – Please label your axes. The titles "Red channel values" should probably be axes labels.

6 – The bars are very confusing and there is no scale given for the gas concentrations. Instead maybe use a colour map where colour/intensity corresponds to value? What do the black dots mean? What is the direction of measurement for the "Distance from ALTZ boundary", i.e. are positive values inside and distance is taken normal to some boundary? If so, which and how determined?

7 – (and elsewhere) it would be useful to have the definitions of ALT, AMT, LTZ, XRD, XRF etc. recalled here. Capitalise XRF (L694)

10 – Please state (either here or in the main text) how Rcum is calculated. Please also use proper representations of units in your axes labels (e.g. "W/sqm")

11 – Please label the abscissa in D and give units. Please also use proper units for ordonate label ("kW/sqm").

Generally, there is no description of the figures in the text of the style "In Figure X we show…", rather the figures are referred to *en passant* (e.g. "pixels of Type 1 to 4 surface show a general increase of mean pixel temperatures from Type 4 to Type 1 surface by an average of 2 degrees (Figure 5).", L330-332). This may be a deliberate stylistic choice by the authors but it this makes it harder still to understand the figures, and leads to the impression that they are not very important, particularly given the paucity of the descriptions in the captions.

---

## Author Comment (AC1)

**Reply letter**

Please find below all the comments made by the Reviewers, followed in each case by our reply in blue colour.

**Reviewer 1:**

Please find below my report on the manuscript "Anatomy of a fumarole field; drone remote sensing and petrological approaches reveal the degassing and alteration structure at La Fossa cone, Vulcano Island, Italy" (egusphere-2023-1692) by Daniel Müller et al.

The present manuscript presents a combination of visible and thermal imagery, mineral and geochemical analyses of rock samples and CO2 soil degassing data to classify and quantify the alteration and degassing structures at the Vulcano Fossa volcano. The manuscript uses a number of novel techniques, with the main emphasis on the classification of thermal and visible images (orthophotos), similar to previously published work (Müller et al., 2021), with the addition of mineralogical and geochemical analyses of hydrothermally altered samples. Whilst this approach is promising, and the results given are interesting, I think that some of the methods employed need much more extensive descriptions. Some of the techniques employed may also be flawed or, at best, misunderstood.

**Reply:** We appreciate this feedback and made all the requested additions and clarified the techniques used. We hope to have a much-improved manuscript. Please find the responses to your suggestions and the changes applied to the manuscript in detail below.

**Reviewer 1:** Lastly, I do not feel that the authorship is currently up to standard for publication, though the written English is fine in itself. Hence it is my opinion that the manuscript requires major revisions before it could be considered for publication. I provide some suggestions that the authors may chose to follow. Given that the revisions may be substatial, a review of the revised manuscript may be necessary. In this case I would be happy to act as referee should you require my services.

**Reply:** We appreciate this constructive review and made many changes that helped to sharpen our manuscript including the method section. Further, we highlight the findings of the suggested works and now refer to them, especially in the discussion section. For more details, please see our replies below.

**Reviewer 1:** PCA and image classification

Probably my major concern regarding the manuscript concerns the lack of description of the "Principal Component Analysis", (PCA) and the image classifications. My reasons are twofold:

The PCA and image classification processes, provided within the ArcGIS propitiatory software, are "black boxes" with no description of what is happening under the hood. As written, it is not easy to understand how these process work, or what data they provide. The interested (and reasonably competent) reader should be able to reproduce the results of this study and I don't think that is currently the case.

In order to reproduce these results, the interested reader would require access to the proprietary ArgGIS software which means paying 500-700€. I don't think that this is fair. Whilst

this may not be an issue for wealthy universities, it could block researchers with less access to funding from utilising the approaches employed in the present study. A proper description of the methods could allow such researchers to look for cheaper of even
open-source alternatives.

**Reply:** We agree and now provide a more detailed description of the methods allowing researchers to reproduce results and/or apply open-source alternatives. We also have added more information on how PCA in the image analysis toolbox of ArcGIS works and outline how the procedure in this toolbox differs slightly from other algorithms. We have moreover added citations to publications that describe the principles of PCA in some detail so that the interested reader can learn more about this approach.
In particular, we have modified and expanded Section 3 'Data and methods' with respect to the steps applied by us to achieve the results shown. This includes all relevant information on threshold values chosen and sub-data sets produced. We hope to have now sufficiently clarified our workflow considering the anomaly detection.

**Reviewer 1:** PCA consists of taking a multi-dimensional dataset and finding the orthonormal basis vector space that describes the data whilst minimising the variance along each vector basis (component). This is achieved by calculating by projection of the data onto a set of orthogonal axes where the variance of each data set is represented by the eigenvalues of the data. These eigenvalues are the principal components. This is usually achieved using a reduced singular value decomposition (SVD) which produces the eigenvalues of the dataset. PCA then takes the list of ordered eigenvalues which are typically used to perform dimensional reduction in high-dimensional data sets. This consists of rejecting any components that do not contribute significantly to the overall variance of the data.
In the present study this is applied to an RGB image (i.e. 3D) and the authors take 3 principal components, so there is no dimensional reduction and hence the "PCA" is kind of redundant.

**Reply:** We modified the method section and hope to now better explain how PCA provides value to this study and image data of volcanoes in general.
It is right that PCA can be used for dimensionality reduction of multidimensional data like multi or hyperspectral data in the sense that it provides the most significant information in a few Principal Components omitting redundant or insignificant information and therewith contributes to a reduction from multiple dimensions to a few that are significant.
However, the way we use the dimensionality reduction of PCA differs a bit from that.
In the first order, we use PCA to obtain variance representations as those highlight anomalies obscured in the original data of correlated RGB bands.
But we also use single Principle Components as a form of "digital scissor" allowing us to constrain areas of interest based on a single PC value (instead of 3 correlated RGB bands) and to create clipping masks to extract the respective data of interest from the original RGB data in a clean pixel based way. Considering the high resolution of the data and the tens to hundreds of millions of pixels we are dealing with, this is very beneficial for obtaining clean data sets for further analysis. So we use the dimensionality reduction just as a side effect for obtaining clean sub-data sets based on a single band clipping, which we investigate then

further. This is now outlined in the revised manuscript and we hope with our modified method section this becomes more clear.

**Reviewer 1**: Regardless of how we name this process, it is unclear what the ArcGIS algorithm produces in the "Principal Component" band images (cf. L220-223) – PCA gives only the variance. Please indicate how this information is used to transform the RGB image.

**Reply:** We agree and have modified the method section 3.2. We now provide a more detailed description of the PCA and added references for principles and applications of PCA for the interested reader. The modified section concerning the general principles and benefits of PCA can be found along lines 224-238 and a section with more detailed steps and threshold values of our analysis can be found in lines 239-253

**Lines 224-238 (general description):** "PCA is a statistical tool that was invented by Pearson (1901), further developed and widely applied in remote sensing or image analysis (e.g. Loughlin, 1991; Fauvel et al., 2009; Alexandris et al., 2017). It can detect and highlight optical anomalies within an RGB data set by transforming the data values of the initial RGB channels onto their perpendicular axes of the highest data variance (e.g. Abdi & Williams, 2010). This can be achieved in several ways. We used the PCA implemented in the ArcGIS image analysis toolbox (see ArcGIS online documentation for Principal Component Analysis), performing the following workflow. In the first step, an ellipse including all data points is calculated for each dimension (RGB). The main axes of these ellipses represent the Eigenvectors (direction of highest variance) and will be used as a new coordinate system for the data transformation. By transforming all data points onto this new coordinate system, we obtain Principal Components (PC) which are variance representations of the initial RGB image data and can be used to detect and highlight optical anomalies like color changes due to alteration processes (Müller et al., 2021, Darmawan et al., 2022). PCA further promotes a decorrelation of the initial RGB bands, a dimensionality reduction, and associated better data separability so that color variations, before expressed by changes in the three RGB bands (3-dimensional problem), can now be accessed in single bands, the single Principal Components (PC). While Principal Component 1 (PC) resembles ~91.3 % (95) of the initial data variance, it mainly shows brightness changes within the image. PC 2 and 3 contain 7.4 (4.5) and 2.3 (0.5) % of the data variance, resemble color changes, and are suitable to resolve optical anomalies related to hydrothermal alteration."

**Lines 239-253 (Details of own processing steps):** "In our data, hydrothermally altered areas were defined based on the PC3, with pixel values > 85 representing hydrothermal alteration. We used this as a mask to crop the respective pixel locations from the original orthomosaic (RGB), resulting in a 16 Mio pixel alteration raster subset (RGB). This alteration raster subset allows for a more sensitive image analysis due to the reduced spectral range with respect to the original orthomosaic. Another iteration of PCA, now applied to the extracted alteration raster subset adjusts to the new reduced spectral range, as we are excluding all redundant data e.g. unaltered surface, and provide a variance representation of the altered surface exclusively. We classified the result in an unsupervised classification (implemented in ArcGIS,

using a combination of Iso Cluster and Maximum Likelihood classification) with 32 classes. We decided for unsupervised classification as this is a more data-explorative way of exploring the pixel information rather than classifying based on a spectral range constrained by training areas defined on pre-assumptions. The 32 classes are chosen to obtain the best possible class resolution, as this is the highest number of classes possible in the unsupervised classification tool. By combining these classes in a way that they resemble larger optical spatial units, we eventually defined 3 Types of alteration surface (Types 1-3) and unaltered surface (Type 4) and further analyzed their spectral characteristics and spatial distribution. Boxplots of the distribution of RGB values in the 32 classes and the spectral range of Type 1-4 surfaces are shown in Appendix A. The optical structure of the fumarole field and alteration zone is similar to the thermal structure and will be discussed in Chapter 4.3.

In case a more extensive description is wished, this could be provided but then should be shifted to the supplementary section.

**Reviewer 1:** The image classification, named as "unsupervised classification", process is poorly documented. Indeed unsupervised classification is a blanket term for a multitude of different families of algorithms so, to be able to reproduce these results, one would have to know which algorithm was chosen and why. The choice of 32 classes seems to be completely arbitrary (50 were used in Müller et al., 2021) and, furthermore, the individual classes are then regrouped (see fig. A1 for example). Could a smaller set of classes (say 4) not have been used to obtain similar results? I strongly urge the authors to justify their choice.

**Reply:** We appreciate this comment and have added further descriptions on the choice of the classification algorithm and the number of classes. See replies to the comment above. The changes applied can now be found in lines 246-253.
We further note that classifying is a subjective process to a certain degree. However, with the data sets provided, descriptions of the tools and parameters used, and the information in Appendix A1, the results should be well reproducible. We hope that in our edited method section the workflow now becomes clearer.

**Reviewer 1:** Whilst the authors refer to previous work (Müller et al., 2021) as a source for their methods, but neither the PCA nor the classification strategies are sufficiently well described in that work either. Without having to detail the algorithms in their entirety, the authors should please sufficiently explain their methods in the present manuscript so that they can be followed with the aim of reproducing their results.
**Reply:** We appreciate this comment and reviewed and expanded the method section with more detail on the methods used. We hope to have improved the manuscript now in a way that the reader can follow the workflow applied and reproduce the results obtained. Please see the replies to the comments above.

**Reviewer 1:**
**Thermal image processing**

F ig. 1: "Brightness" temperature predicted by Planck's law compared to the scaling proposed by the authors (eq. 1). The two curves cross at a temperature of 383 K (about 90°C). The authors state that they produce temperature maps from the 16-bit radiometric greyscale orthophoto using a linear mapping given by their eq. 1 using the radiometric resolution as a scaling factor. Owing to the non-linear behaviour of IR sensors this would seem unlikely to hold for more than a very limited range of greyscale values. Furthermore, it is typically necessary to use Planck's law which predicts the "brightness" temperature of an object from the intensity of incoming radiation registered by the sensor (i.e. radiometric value). By way of illustration, I have produced the above graphic (Fig. 1). Here we see that there is only one point of intersection for the two curves and the predicted temperatures can be drastically different. That said, I am not familiar with the FLIR Tau camera as used in this study and do not have access to the radiometric conversion factors necessary to correctly plot the curve for this camera. If it is like many other FLIR and other IR cameras that I have used, this information can be found in the EXIF (image metadata) which can be readily extracted using the ExifTool software, for example. However, the authors should check their data and any calculations that depend on the temperature. Some of the stated temperatures and thresholds (e.g. 40 °C for the "high-temperature fumaroles") are rather low given the vent temperatures recorded in other works (vent temperatures are well in excess of 100°C, cf. Diliberto, 2017; Mannini et al., 2019; Diliberto, 2021).

**Reply:** We appreciate this comment. However, the formula used is recommended by the camera provider for models like Vue or Tau. We tested the results of the calculated temperature values and compared them with temperatures shown for the original data in the Thermoviewer software on many pixels and found that temperatures accurately match.
The apparently low-temperature values can be due to several factors. Referring to Figure 4 D, for instance, we are only showing the lower threshold of 40°C as a value, but the high-temperature fumarole zone labeled as HTF would include temperatures of up to ~160°C. The range of temperatures observed is described earlier in the methods section line 280.
However, apart from that, we flew at an altitude of ~150 m above the fumarole field what results in a pixel size of ~40x40 cm on the ground. Very high temperatures at the surface appear typically very localized so we deal with in part mixed information in some pixels, i.e. the localized high temperatures of fumaroles, and the colder surroundings. For this reason, we state in chapter 3.3 line 275 and following that remotely sensed temperatures are only apparent temperatures, sensitive to environmental influence, and can not be compared to in situ values. Temperatures shown by Diliberto (2017 and 2021), for instance, refer to the long-term measured in situ temperatures, that are also measured at larger depths (~30-50 cm) while we sense the surface only.

**Reviewer 1:**
**Authorship and increasing the scope of the paper**
A quick survey of the first dozen or so references showed that they were cited once in the introduction and nowhere else in the manuscript. The discussion contains only six references, and is often a rehash of the results section rather than a forum for putting these results into a fuller context and comparing them to previous works on Vulcano and, potentially, other

volcanoes. Several references (e.g. Chiodini et al., 1996; Chiodini et al., 2005; Harris et al., 2009; Mannini et al., 2019) have made estimations of degassing and/or heat budgets with Harris et al. (2009), in particular, having made detailed descriptions of the fumarole field. I find it strange that the authors have not chosen to make the comparison with these works, particularly given the ongoing and unrest at Vulcano with the recent well-documented paroxysms.

**Reply:** Thank you for this comment. In our revised version we now cite these works and highlight findings made earlier in our discussion. In addition, we discuss our results with heat budgets in the mentioned studies. For this purpose, most of the discussion chapters, but in particular chapter 5.3 were extensively modified.

**Reviewer 1:** Fig. 11 identifies a structure with an increased radiant density and labels it as "new fumarole complex in 2021", but this is not discussed anywhere. This would be very important information for assessing the activity at this volcano.

**Reply:** A new fumarole complex has indeed formed during the 2021 volcanic crisis. We agree that this is very interesting, and added this to the discussion in sections 5.2 and 5.3. However, as we are also working on another publication that in detail considers changes during the crisis based on drone-based infrared and optical time series, we would refrain from going into too much detail in the framework of the current study.

**Reviewer 1:** Curiously, Fig. 10 hints at a heat budget having being calculated, but this is not discussed in the manuscript.

**Reply:** We appreciate this comment. We added an additional sentence to the methods section to explain better our radiation calculations. The respective section now reads:
"To compare the thermal emissions… To compare identified units quantitatively, we summarized the radiation per pixel for the respective units a-g to cumulative thermal radiation (Rcum). "

**Reviewer 1:** I note also that the area of the alteration zone (ALTZ), that is the area affected by hydrothermal activity, is given as 70 000 m2 (note typo "770 000 m2" on L301) which is very close to the "diffuse heated area" of 63 000 m2 calculated by Mannini et al. (2019) using approximately contemporaneous data. Of course, I have my own professional biases in mentioning this, but
the authors have already done the work so it is suprising that it is only mentioned in passing.

**Reply:** We agree with the reviewer. We have now fixed the typo and we have further highlighted the work of Mannini et al. 2019 in our revised manuscript. We detailed the distinction between vent flux and diffuse flux zone made by them and discussed how our work can contribute to the previous studies. The respective section can be found in lines 543 and the following and reads:

"Alteration effects can be observed in an area (ALTZ) that is actually ~50 times larger, and a thermally active surface that is ~25 times larger than the area covered by high-temperature fumarole complexes. The ALTZ is similar to the diffuse flux zone defined by Mannini et al. (2019). However, within this zone, we observe variability that represents local alteration gradients or structural units. Therefore, we can add detail to the surface structure and activity pattern identified in previous works, e.g. Mannini et al. (2019) who divide the active region into a vent flux zone and a diffuse flux zone which are often used as reference, or Harris et al. (2009). In particular, we can contribute detail to the vent flux zone by outlining high-temperature fumarole locations based on high-temperature pixels. Further, we can show that thermal radiation and gas flux of the ALTZ or diffuse flux zone is not uniform but shows a strong local variability with high fluxes in identified active units a-g and low fluxes in larger parts of the central fumarole field, then associated with LTZ (Low-Temperature Zones)."

**Reviewer 1:** Concerning alteration of the edifice, there have been a number of studies in recent years trying to ascribe thermal properties of volcanic rocks to hydrothermally altered samples (typically andesite). These results may be interesting to discuss. See Heap et al., (2022) in particular. Section 5.3 briefly mentions the role of alteration and permeability: "Relative gas flux values measured within unit c are lower than observed for units a and b, for instance. This might be a consequence of the dynamics of hydrothermal alteration and indicate permeability reduction or sealing processes due to the advanced state of alteration like proposed by Heap et al., 2019." (L592-594) Furthermore, a sequence of alternating high and low permeability zones are identified in Fig. 11, but each result is only discussed independently and there is no real synthesis of the large and important set of results. This is one discussion point, in particular, that is really important for assessing volcanic hazard and I find it frustrating that this point has not been fully persued.

**Reply:** We accept the criticism. We have now modified all chapters of the discussion section to a more comprehensive summary of the identified structures and their implications, better highlighting the relevance of the single units for the degassing system. We now provide a better synthesis of our results.

**Reviewer 1:** A minima, the figure captions should allow the reader to understand the figure in isolation and, currently, this is not the case for several of the figures. The main cuprits are

**Reply:** We appreciate this comment and modified most figure captions to a more extensive description.

**Reviewer 1:** Fig 2 – give locations of each photo, also show these locations on Fig 1B.

**Reply:** We appreciate this comment. We indicated the locations of the field photos in Figure 2 (now in Figure 1 C) and added the respective reference to the figure caption of Figure 1.

**Reviewer 1:** 3 – describe the grey blocks. Why are there two blocks for Remote Sensing?

**Reply:** We appreciate this comment and extended the Figure caption, describing in detail the approach and purpose of the different steps used for this study. The new Figure caption now reads:

"Figure 3: Overview of the general workflow used for this study. An anomaly detection from optical and thermal infrared remote sensing data allows us to reveal the anomaly pattern and infer the surface structure of the degassing and alteration system. To validate the observed structure, the remote sensing study was complemented by surface degassing measurements revealing the present-day degassing pattern, and by X-ray diffraction and X-ray fluorescence analysis of selected rock samples to prove different alteration units based on changes in mineralogical and bulk-chemical composition. Continuous monitoring using high-resolution remote sensing data also makes it possible to record dynamics within the system and draw conclusions about the general condition of the degassing/alteration units, e.g. with regard to surface sealing. "

**Reviewer 1:** 8 – what are "RGB values", as only one value is given here. Is this for one band in particular or an average? Why are RGB values being used rather than one of the "PCA" image bands?

**Reply:** We appreciate this comment and agree that the plot is not described sufficiently. To address this, we modified the Figure caption and added all relevant information for understanding the plot. The new figure caption now reads:

Figure 8: Relation of inferred surface type (T1-4) and sulfur content of rock samples taken in the respective surface T1-4. Black dots mark the sulfur contents of rock samples and are shown on a log scale versus the surface type from which the sample was taken, labeled by Type 1-4 on the x-axis and demarked with boxes using the same color code as throughout the manuscript. From Type 1 surface to Type 4 surface we see a significant decrease of sulfur content from up to 100% for Type 1 to <1% for Type 4 surface. The black dashed line illustrates this. An exemption is the Type 3 surface where we observe two distinct clusters, one with low sulfur values and one with exceptionally high sulfur values. These high sulfur values belong to samples taken in the LTZ (low-temperature zones), which separate fumaroles from the larger diffuse active units a and b (compare Figure 4) for instance. These samples indicate that LTZ represents sulfur-rich crusts that block heat and gas flux from the surface. The colored boxplots show the distribution of RGB values for the respective surface Type 1-4, showing a similar trend of decreasing values. Red boxplots represent the red value, blue and green the respective blue and green channel values of the image. The coincidence between both indicates a direct relation or control of sulfur onto the surface colorization.

**Reviewer 1:** 9 – What is the absicssa in this figure? How is it that Transect A blends into Transect B? I think it would be worthwhile to combine this figure with
**Reviewer 1:** Fig. 7, particularly as one requires the locations of the transects to understand what is going on in this figure.

**Reply:** We appreciate this comment and made the changes suggested. We added the axis labels, split transect A visually from transect b, modified and expanded the figure caption, and referred to Figure 7 for the locations of the transects and single samples. The figure should now be easier to understand.

**Reviewer 1:** Other suggestions for the figure captions are:
4 – D, please give the thermal thresholds for each class.

**Reply:** We agree and added the thermal thresholds for high-temperature fumaroles and diffuse active units to Figure 4D.

**Reviewer 1:** 5 – Please label your axes. The titles "Red channel values" should probably be axes labels.

**Reply:** We appreciate this comment and add the respective axes labels to the figure as well as an expanded description to the figure caption which now reads:

"Figure 5: Boxplots of RGB color value- and temperature distributions observed for the different surface Types 1-4 (T1-T4), selected areas a-g, and associated low-temperature zones LTZ1-3. Surface types and locations of identified units are depicted in Figure 4B/D. A) Red channel value distribution. B) Green channel value distribution. C) Blue channel value distribution. D) Temperature value distribution. Values are based on an analysis of 6.8 million pixels within the ALTZ."

**Reviewer 1:** 6 – The bars are very confusing and there is no scale given for the gas concentrations. Instead maybe use a colour map where colour/intensity corresponds to value? What do the black dots mean? What is the direction of measurement for the "Distance from ALTZ boundary", i.e. are positive values inside and distance is taken normal to some boundary? If so, which and how determined?

**Reply:** We appreciate this comment and modified the Figure and expanded the caption with a more detailed description. In detail, we add the flux values to the legends in A and B, a dashed line representing the ALTZ outline in Figures C and D, and adapted font sizes. We expanded the caption to :

Figure 6: A/B) Spatial distribution and flux values for $CO_2$ (red bars in A), $SO_2$ (light blue bars in B), and $H_2S$ (turquoise bars in B) in a map view for 200 measurement points. Each bar represents a relative flux value at a measurement location. In case no flux was detectable, the respective location is marked with a black dot only. The dashed line highlights the ALTZ (Alteration Zone) boundary. Dark grey features in the background highlight the thermally active surface (compare Figure 2D). Labels F0, F5AT, F11, and FA mark prominent high-temperature fumaroles. The labels a-g mark notable large-scale anomaly units. C/D) Flux values are plotted by distance to the ALTZ boundary (dashed line). Measurement points within the ALTZ are represented by positive distances from the ALTZ boundary (highlighted by blue background)

and measurements outside the ALTZ by negative distances (unaltered). A generally higher flux is observed within the ALTZ, but while $CO_2$ is also abundant outside the ALTZ, significant $SO_2$ and $H_2S$ fluxes were observed exclusively within the ALTZ, especially on the outer edges and associated with units a-g. The averaged flux values (av) are depicted in the respective sections of C and D.  E-G) Relative flux values of identified units (a-g), Type 3 surface (T3), the unaltered surface (un), and fumaroles on the northern rim at a distance (of) highlight the spatial variation of different gas species with high flux values for units a and b, lower flux values for e.g. unit c and generally lower flux of sulfuric gas species in the unaltered regime outside the ALTZ.

We hope our improvements now allow a better understanding of the figure. The revised figure now addresses most of the suggestions. However, we would prefer to stick to the bar plots in Figure A/B as after assessing many different versions of this figure, we found the current version to be the best representation. With the added flux values and descriptions in the caption, the figure should now be better understood.

**Reviewer 1:** 7 – (and elsewhere) it would be useful to have the definitions of ALT, AMT, LTZ, XRD, XRF, etc. recalled here. Capitalise XRF (L694)

**Reply:** We appreciate this comment and modified the figure and caption, including definitions for the mentioned abbreviations. We capitalized XRF as suggested. The new figure caption now reads:

Figure 7: Mineral and bulk chemical composition of rock samples along 3 transects A-C, crosscutting alteration gradients, and structural units. A) Overview map with defined surface types 1-4, highlighting the area with visible optical changes at the surface, referred to as ALTZ (Alteration Zone, marked by a dashed line). The three transects A-C were placed so that they crosscut prominent units. Mineralogical compositions from XRD (X-ray diffraction) are depicted by circular plots at the bottom of Fig. A.  B) Bulk chemical composition from XRF (X-ray fluorescence) analysis of transects A-C. Transect A/B) With increasing alteration intensity, we observe a relative decrease of the initial mineral phases sanidine and cristobalite whereas the sulfur content increases. Note that the mineral composition in this figure is normalized to 100 % non-amorphous minerals. In the chemical composition, we observe a significant decrease of $Al_2O_3$ and $Fe_2O_3$ but an increase of MnO, $TiO_2$, and S with increasing alteration, especially at the Type3-Type1 (T3-T1) boundary marked in light blue. For transect C, we observe a dominant increase of S (17-40%) for samples taken within the LTZ (low-temperature zone). Compared to the other transects, changes in $Al_2O_3$, $Fe_2O_3$, $TiO_2$, and MnO are less significant. For samples marked with an asterisk (*) XRF results are not available. Also, no XRD results are available for transect C.

**Reviewer 1:** 10 – Please state (either here or in the main text) how Rcum is calculated. Please also use proper representations of units in your axes labels (e.g. "W/sqm")

**Reply:** We appreciate this comment and modified the figure caption and method section accordingly. We now define that Rcum is the cumulated radiant exitance of thermal pixels associated with the identified active units. The new caption reads:

Figure 10: Anatomy of the fumarole field. A) Simplified structure of the fumarole field highlighting surface types and structural units of increased diffuse activity (a-g) or areas of apparent surface sealing (LTZ1-3 marked by white lines). B) Contribution in % to the total radiation for HTF (high-temperature fumaroles), units a-g, and LTZ1-3 (low-temperature zones) considering pixels with T>22°C and for identified units based on a spatial constraint, also including temperatures <22 °C. C) Radiant exitance (RE) in W/m² and cumulative radiation (Rcum) in MW. Rcum is the cumulated radiant exitance (Equation 2 in the method section) of all pixels associated with the respective active unit. Note that for Rcum only pixels with T>22°C were used. We can clearly distinguish different thermal regimes that are also coincident with surface types identified in the optical data.

**Reviewer 1:** 11 – Please label the abscissa in D and give units. Please also use proper units for ordonate label ("kW/sqm").

**Reply:** We appreciate this comment and modified the figure as suggested.

**Reviewer 1:** Generally, there is no description of the figures in the text of the style "In Figure X we show…", rather the figures are referred to en passant (e.g. "pixels of Type 1 to 4 surface show a general increase of mean pixel temperatures from Type 4 to Type 1 surface by an average of 2 degrees (Figure 5).", L330-332). This may be a deliberate stylistic choice by the authors but it this makes it harder still to understand the figures, and leads to the impression that they are not very important, particularly given the paucity of the descriptions in the captions.

**Reply:** We appreciate this comment and agree. Some of the captions were kept too short with important information still needing to be included. We modified most of the figure captions now and added more detailed descriptions. We addressed each of your above-mentioned points and hope the figures are now easier to understand.

---

## Author Comment (AC2)

**Reviewer 2**

**Reviewer 2:** The paper presents a comprehensive and meticulous investigation into the degassing and alteration structures of the fumarole field at La Fossa cone on Vulcano, offering valuable insights into the complex dynamics of volcanic activity. Using innovative methodologies, including close-range remote sensing, mineralogical and geochemical analyses, and surface degassing measurements, the authors provide a detailed and multi-faceted examination of the degassing system.

One of the study's most commendable aspects is its integration of high-resolution drone-derived imagery with traditional analytical techniques. This approach enables a nuanced spatial analysis, allowing the authors to accurately identify and characterize major active units. Furthermore, the quantification of thermal energy release provides valuable quantitative data on the relative importance of different degassing features within the system, enhancing our understanding of volcanic processes.

The authors demonstrate a commendable level of transparency and rigour by acknowledging potential limitations, such as gas plume distortion affecting image quality. This ensures the reliability and validity of their findings, reflecting a commitment to scientific integrity and strengthening the credibility of the study's conclusions.

Overall, this paper represents a significant contribution to the field of volcanic degassing research. By elucidating the complex interplay between surface manifestations, alteration gradients, and gas emissions, the study advances our understanding of volcanic systems and provides a solid foundation for future research in this area. Its comprehensive approach and meticulous attention to detail make it a valuable resource for scientists and researchers working in volcano monitoring and hazard assessment.

**Reply:** We appreciate this positive feedback and thank you for reviewing our manuscript. We addressed all your suggestions and modified our manuscript accordingly. Please find the detailed replies to each of your suggestions below.

**Reviewer 2:** Here I list minor comments that I hope may improve the final version of the manuscript:
- The authors mention some fumaroles (e.g., F0) but do not show them in the figures in the main text (only supplementary). They are not also mentioned in the discussion. The fumaroles F0, FA, F5AT and F11 have distinct features that could correlate with the paper's findings. I recommend reading Aiuppa et al. 2006 GRL and Tamburello et al. 2011 JVGR. These historical fumaroles should be plotted at least in Fig. 1 and 6.

**Reply:** We agree that showing the locations of fumaroles could help to better orient in the Figures and added the locations of fumaroles F0, F5AT, F11, and FA to Figure 1 and Figure 6, and the respective description in the Figure captions. Further, we added the labels for relative flux values to Figure 6A/B.

**Reviewer 2:** Authors describe the colour of the 4 different surfaces. I suggest that it could be more straightforward to show these colours (or a palette of colours for each type) in one of the figures;

**Reply:** We appreciate this comment. We intended to show color samples for surface types 1-4 in Figure 4 below Figures A and B as small subfigures. We understand that they were too small for easy viewing. We have now increased the size of these color samples and added descriptions to the figure caption.

**Reviewer 2:** Please describe in Figure 4 caption what the letters a-g are;

**Reply:** We agree and now better highlight in the Figure caption what the labels a-g mean. The respective sentence now reads "The labels a-g demark notable large-scale anomaly units that can be observed in both, the optical and the thermal data."

**Reviewer 2:** The bars in Figure 6a-b are hard to read. I suggest to use coloured circles with a colour bar. I suggest to calculate also the ratio between fluxes ($CO_2/S_{tot}$, where $S_{tot}$ = $SO_2$ + $H_2S$) and to plot their distribution to highlight the role of sulfuric gases

**Reply:** We appreciate this comment. However, we tried different versions of this figure before but concluded that the bar plot was the best representation for showing the relative fluxes. Calculating the ratios from these measurements would certainly be interesting. However, we feel that calculating ratios from our results is not optimal due to the differential uncertainties of the instrument applied. We would feel a need to compare measured ratios to real in-situ data first. We have more gas data from other years and will look into that in more detail, but note that this is beyond the scope of the present study.

**Reviewer 2:** 150 Fumarolic temperature rose up to 690 °C in May 1993 (Chiodini et al., 1995) Chiodini G., Cioni R., Marini L. and Panichi C. (1995) Origin of fumarolic fluids of Vulcano Island, Italy and implications for volcanic surveillance. Bull. Volcanol. 57, 99–110. http://dx.doi.org/10.1007/BF00301400

**Reply:** We appreciate this comment and modified the text. It now reads: "Gases of the high-temperature fumaroles (HTF) emerge with temperatures >300 °C, but temperatures have been exceeded during previous volcanic crises (Harris et al., 2012, Diliberto, 2017). Temperatures of up to 690 °C were reported from May 1993 by Chiodini et al. (1995)."

**Reviewer 2:** 240 Please explain how the 40°C threshold has been chosen;

**Reply:** We appreciate this comment and have now defined thermal units and temperature thresholds in more detail.
Specifically, the temperature thresholds were defined after analyzing our infrared and optical data as well as based on previous knowledge of fumarole locations from previous field campaigns. When classifying pixels based on their temperatures, they form spatial clusters. We

found that the 40°C temperature threshold outlines well the physical locations (depressions or fracture-like shapes in the surrounding fumarole crust) of major high-temperature fumaroles.

The 22-40°C threshold defines larger contiguous clusters, which we interpret as rather diffuse features. However, these assumptions are partly arbitrary but a necessary approximation in order to define spatial boundaries and be able to quantify thermal emissions, size and extent of different active units.

In the text, we changed lines 240 and the following to: "The temperature map was used to define the thermal structure. We observed several distinct thermal spatial units with temperatures significantly above the background temperature, that can be distinguished in high-temperature fumaroles (HTF in the following) and areas of rather diffuse thermal surface heating (Figure 4 B/D). To constrain these units spatially for further comparison, we had to approximate spatial boundaries what was done after comparison to our optical data and based on knowledge of previous observations by defining the temperature thresholds of T = 22-40°C for the diffuse heated areas and T> 40°C for HTF. The 40°C threshold resembles well the known locations and extent of HTF in the upper fumarole field."

**Reviewer 2:** 530 Also, halogen may play a role in chemical leaching (Aiuppa et al., 2009 Chem Geo);

**Reply:** Yes we agree with this statement. We cannot constrain this process with our data set but have now added a reference to this possibility.

**Reviewer 2:** 543 "higher gas flux" should be "higher acid gas fluxes"?

**Reply:** We agree and modified the phrase to "… a higher acid gas flux…" as suggested.

**Reviewer 2:** 544 "we observe similar" looks incomplete

**Reply:** We appreciate this comment and changed the respective sentence to the following wording: "Analyzing the broader area of the central crater region we can infer multiple other areas where we observe similar changes of colorization indicating similar argillic or strong silicic alteration effects at the surface, in particular, located on the southern inner crater, the outer crater rims, the 1988 landslide area (Madonia et al. 2019) and the northern flank towards Vulcano Porto. These zones of strong alteration are indicated in red in Figure 1B.

---

## Author Response (AR2)

Dear Editor

We have gone through the manuscript thoroughly, made the last changes, removed typos and updated the last figures. The uploaded version "proofread_manuscript" should now meet the journal standards and be ready for publication. Finally, we have also added a link to a data repository where the remote sensing data of the study can be accessed.

Thank you very much for your support and handling of our manuscript.

Yours sincerely

On behalf of all co-authors

Daniel Müller